# Unveiling charge utilization mechanisms in ferroelectric for water splitting

Jie Zhang[1,2], Yong Liu [1], Thomas Dittrich[3], Zhuan Wang [4], Pengxiang Ji[2,5], Mingrun Li[1], Na Ta [1], Hongyan Zhang[6], Chao Zhen [7], Yanjun Xu[4], Dongfeng Li[1,2], Zhendong Feng[1,2], Zheng Li [1], Yaling Luo[1,2], Junhao Cui[1,2], Dong Su [5], Yuxiang Weng[4], Gang Liu [7], Xiuli Wang [1], Fengtao Fan [1] ✉ & Can Li [1,2] ✉

Charge separation is a critical process for achieving high photocatalytic efficiency, and ferroelectrics hold significant potential for facilitating effective charge separation. However, few studies have demonstrated substantial photocatalytic activity in these materials. In this study, we demonstrate that in ferroelectric $PbTiO_3$, surface Ti vacancy defects near the positively polarized facets impede photocatalytic performance by trapping electrons and inducing their recombination. To tackle this issue, we selectively grew $SrTiO_3$ nanolayers on the polarized facets $PbTiO_3$, effectively mitigating interface Ti defects. This modification establishes a efficient electron transfer pathway at the interface between the positively polarized facets and the cocatalyst, extending the electron lifetime from 50 microseconds to the millisecond scale and significantly increasing electron participation in water-splitting reactions. Consequently, the apparent quantum yield for overall water splitting achieves the highest values reported to date for ferroelectric photocatalytic materials. This work provides an effective strategy for designing advanced ferroelectric photocatalytic systems.

Since the discovery of ferroelectrics in Rochelle salt in the 1920s[1], ferroelectric materials have garnered considerable attention due to their unique structural properties at the unit cell level for various applications. The symmetry-breaking nature of the electrostatic boundary condition, which generates a permanent depolarization field confers ferroelectrics with unique physical properties. Presently, studies on ferroelectrics mainly fall into two categories. One focuses on polarization variation, such as ferroelectric memories[2,3], negative capacitance[4,5], and the electrocaloric effect[6,7]. The other explores the

coupling between ferroelectric polarization and light, such as the electro-optic (non-linear optics) effect[8], bulk photovoltaic effect (BPVE)[9–13], and photoelectric or photocatalytic applications[14–16]. Despite significant efforts in both areas, the practical application of ferroelectrics still faces significant challenges.

One of the most promising and challenging applications of ferroelectrics is the conversion of solar energy into electricity or chemicals, facilitated by the intrinsic advantages of asymmetrical unit cells for charge separation and transfer[17,18]. Distinct from other

[1]State Key Laboratory of Catalysis, Dalian National Laboratory for Clean Energy, The Collaborative Innovation Centre of Chemistry for Energy Materials (iChEM), Dalian Institute of Chemical Physics, Chinese Academy of Sciences, Dalian 116023, China. [2]University of Chinese Academy of Sciences, Beijing 100049, China. [3]Helmholtz-Zentrum Berlin für Materialien und Energie GmbH, Institut für Silizium-Photovoltaik, Kekuléstr. 5, Berlin 12489, Germany. [4]Laboratory of Soft Matter Physics, Beijing National Laboratory for Condensed Matter Physics, Institute of Physics, Chinese Academy of Sciences, Beijing 100190, China. [5]Beijing National Laboratory for Condensed Matter Physics, Institute of Physics, Chinese Academy of Sciences, Beijing 100190, China. [6]Institutional Center for Shared Technologies and Facilities, Dalian Institute of Chemical Physics, Chinese Academy of Sciences, Dalian 116023, China. [7]Shenyang National Laboratory for Materials Science, Institute of Metal Research, Chinese Academy of Sciences, 72 Wenhua Road, Shenyang 110016, China. ✉e-mail: ftfan@dicp.ac.cn; canli@dicp.ac.cn

semiconductors, where carrier migration primarily occurs through drift and diffusion, ferroelectrics exhibit a unique mechanism: before screening, carriers predominantly migrate by drifting within the depolarization electric field. Specifically, positive holes and negative electrons move in opposite polarization directions[19–22], and increasing the thickness along the polarization direction of ferroelectric PbTiO$_3$ enhances the driving force for charge separation, consequently augmenting the photocatalytic hydrogen evolution activities[19]. This distinct structural characteristic of ferroelectrics makes them particularly suitable as photocatalysts.

Despite the theoretical threshold of around $10^5$ kV/cm for electric field present within ferroelectrics, which is about 3–4 orders of magnitude higher than that measured in conventional semiconductor photocatalysts[23], ferroelectric catalysts have unexpectedly shown relatively poor water splitting performance in actual practice. In fact, the efficiencies reported so far for ferroelectrics are significantly lower compared to other strategies, such as p-n junctions[24–26], heterojunctions[27–31], facet engineering[32,33], defect introduction and doping methods[34–36], cocatalysts[37,38], the Dember effect[39], and Z-scheme systems[40,41].

Efforts have been made to enhance the depolarization field of ferroelectric catalysts by mitigating the internal screening charges, which naturally occur alongside the formation of the crystal structure. In this context, pioneering research has shown that the activity is moderately increased when these internal screening charges are partially eliminated, indicating that such elimination can contribute to improved activity[42]. In particular, a recent study has demonstrated that the growth of SrTiO$_3$ (STO) nanolayers on the positively polarized surface of PbTiO$_3$ (PTO) particles enhances the overall water-splitting activity by a factor of 15[43]. It is hypothesized that the enhancement may be attributed to the reduced Schottky barrier height and increased dispersion of the cocatalyst.

However, despite these significant advancements over the last several decades, it is worth noting that ferroelectrics have a larger electric field for charge separation compared to the other strategies mentioned above[19,44]. This suggests that the inherent driving force for charge separation within the bulk phase of ferroelectric materials is not an impediment to achieving high photocatalytic activity. This raises an important question: what are the key factors limiting the transfer of photogenerated charges to surface reaction sites? Understanding this challenge is essential for unlocking the full potential of ferroelectric materials to achieve significantly enhanced photocatalytic performance.

In this work, a profound mechanistic study was carried out to investigate the role of defective structures on the efficiency of photogenerated electrons from the positive surface of PTO in the overall water-splitting reaction. Our study shows that the main obstacle for efficient photocatalytic water splitting by PTO is the existence of Ti defects at the positive polarization facets that inhibit efficient electron transfer for hydrogen production. By eliminating abundant Ti defective sites on the positive surface of PTO through the growth of STO nanolayers, AQE@365 nm of the overall water splitting was elevated by 400 times due to the construction of a highly efficient electron transfer route. Precise control of surface defects is a potent strategy for improving ferroelectric photocatalyst performance in water splitting.

## Results and discussion

PTO single crystals with uniform morphology were synthesized using the hydrothermal method. The morphological characteristics of the samples were analyzed by scanning electron microscopy (SEM) (Supplementary Fig. S1a) and atomic force microscopy (AFM) (Supplementary Fig. S1b, c). The particles primarily have a size of 600 nm. To substantiate the ferroelectric nature of the PTO samples, X-ray diffraction (XRD) patterns (Supplementary Fig. S1d) and piezoresponse

force microscopy (PFM) (Fig. 1b–d) were measured. The XRD patterns indicate that PTO samples are in a ferroelectric tetragonal phase validated by a standard PDF card (JCPDS 70-0746). Furthermore, the 180° phase-contrasted data and butterfly-shaped amplitude data presented in Fig. 1e showed the reversibility of the polarization direction, which is characteristic of ferroelectrics.

In pursuit of more efficient charge separation, it is imperative to minimize the presence of domain walls and aggregate various types of defects that serve as recombination centers for photo-generated carriers[45]. High-resolution scanning transmission electron microscopy (HR-STEM) data (Fig. 1a) revealed a uniform distribution of the unit cell within the PTO particles, indicating the formation of a single-domain structure. Such a monodomain structure is expected to minimize the charge recombination centers that are favorable for photocatalytic applications.

The single-domain properties were further confirmed by PFM (Fig.1b, c), a conventional method to illustrate the domain distribution in ferroelectric materials. The homogeneous distribution of the vertical amplitude across the two particles (Fig. 1b) signifies the inherent piezoelectric properties in all nanoparticles of PTO. The nearly identical amplitudes of the two particles in phase images (Fig. 1b) indicate the two particles have similar d$_{33}$ values. In addition, the PFM phase images in Fig. 1c and the magnified image in Fig. 1e do not show any contrast for both single particles, suggesting a homogeneous domain structure. The absence of contrast indicates the absence of domain walls in the particles, affirming their monodomain nature. Furthermore, after annealing at 950 °C, the emergence of an antiparallel domain within a single particle, as indicated by a 180° phase difference (Supplementary Fig. S2), confirmed the monodomain properties of the prepared PTO powders.

Despite the lack of phase contrast within individual particles, the observable phase contrast between the two particles indicates that the two particles have differences in domain directions. Figure 1c illustrates a 180° phase difference between the two particles with polarization in opposite directions, one upward and the other downward. In alignment with early reports[19], these particles exhibit the capability to separate electrons and holes in distinct polarization directions (Supplementary Fig. S3). Spontaneous polarization is generally considered to be the origin of the built-in electric field facilitating charge separation in ferroelectrics. The distinct polarization and uniform unit cells suggest the successful preparation of single-domain ferroelectric PTO particles. The study of such single-domain nanoparticles may provide more insights into charge separation and transfer in ferroelectrics.

Our previous study demonstrated that the carriers in ferroelectric PTO could indeed be separated to opposite surfaces, but the photocatalytic water splitting activity was still quite low[36]. We believe that understanding the charge phenomena near the PTO surface is essential to achieve efficient charge separation and transfer for high photocatalytic activity. Therefore, high-resolution scanning transmission electron microscopy (STEM) was applied to study the surface properties of PTO. Figure 2a, d illustrate the cross-section of PTO near the positive polarization surface. The polarization direction could be determined by the Ti atom shift within the unit cell. As depicted in Fig. 2b, the schematic illustrates the upward polarization of the unit cell in ferroelectric tetragonal PTO, where the Ti atom shifts upward along the z-axis, while the oxygen octahedron shifts downward, generating an upward polarization within the unit cell. In the magnified image in Fig. 2c, the Ti atom is shown to shift upward, indicating an upright polarization direction. Consequently, the surface in Fig. 2a can be confirmed as having a positive polarization direction. In Fig. 2a, the red-dotted rectangle highlights the near-surface region on the positive surface of PTO. In Fig. 2d, the surface and bulk regions were delineated by the red dotted line. The angle of the (101) direction in the bulk region was measured to be 41.56° relative to the x-axis. However, at the surface, this angle shifted to 43.09°, suggesting the presence of

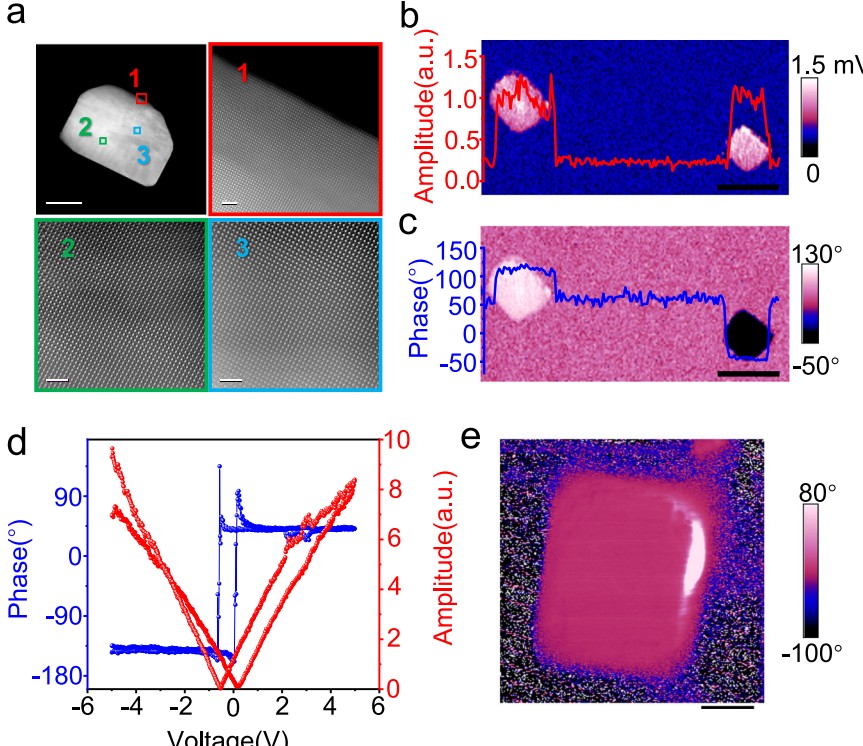

**Fig. 1 | Characterization of monodomain PTO particles. a** SEM and HR-STEM images of PTO particle. The red number 1, green number 2, and blue number 3 correspond to the HR-STEM of the zones magnified as marked in the SEM image with the same-colored labels. **b, c** Images of PTO with different polarization directions showing (**b**) amplitude and (**c**) phase of these two particles, respectively. The red line in (**b**) and the blue line in (**c**) represent the values of the amplitude and phase of the two particles, respectively. **d** PFM hysteresis loop of PTO particles. **e** magnified phase image of a single domain PTO particle. (The scale bars: (**a**) 100 nm (SEM picture at upleft), 1 nm (the other three magnified HR-STEM pictures), (**b**) and (**c**) 700 nm, (**e**) 240 nm, respectively).

distortion in the surface region. To further investigate this distortion, electron energy loss spectroscopy (EELS) of Ti $L_2$ and $L_3$ levels was performed. Supplementary Fig. S4 illustrates the variations in the Ti peaks from the bulk to the surface in PTO, as indicated by the arrow in Supplementary Fig. S4a. Both Ti $L_2$ and $L_3$ levels displayed $e_g$ and $t_{2g}$ peaks due to the splitting of d orbitals of Ti within the oxygen octahedron field. Near the PTO surface, the EELS peaks of $e_g$ and $t_{2g}$ in both Ti $L_3$ levels showed reduced splitting. Such a phenomenon resulted from the distortion of the unit cells in PTO due to the defects on the surface, leading to a reduction in the degeneracy of the d orbitals of Ti within the crystal field of the oxygen octahedron. Both the STEM and EELS results confirm the defective surface structure of PTO. These defects might be relevant to the observed poor activity.

To further study the influence of the surface defects in ferroelectric PTO, a surface-modified ferroelectric-based catalyst was synthesized. This was achieved by selective growth of STO nanolayers on the PTO surface. Meanwhile, the slight increase in absorption below 350 nm indicated that the light absorption properties remained nearly unchanged after STO modification (Supplementary Fig. S6). SEM images in Fig. 3a, b visually confirm the selective growth of STO nanolayers on the PTO surface. The associated energy-dispersive X-ray spectroscopy (EDS) data in Supplementary Fig. S7 supported the assertion that four nonpolar facets were covered. These data indicated that four nonpolar facets and one polar facet are effectively covered with STO nanolayers. Notably, the nonpolar facet exhibits a fully covered film, whereas the polar facet displays pristine particle morphology (refer to Fig. 3b). The growth mechanism of STO on PTO occurred through electrostatic adsorption between the STO precursors and PTO surface, as shown in Supplementary Fig. S8a This mechanism was further confirmed by the selective growth of other materials on PTO (Supplementary Fig. S8b–e).

To identify which polar facet was exposed, a spherical aberration-corrected scanning transmission electron microscopy study was conducted. We selected the interface between the red and blue rectangle regions in Fig. 3b, corresponding to the magnified region marked by the red rectangle in Supplementary Fig. S22. It is evident that the opposing side was not covered by STO. The atomic arrangement across the PTO-STO interface (Fig. 3c) shows a well-structured configuration. The disappearance of distortion is a strong indication of defect reduction at the interface of PTO. The polarization direction of PTO is shown through the magnified image in the upright of Fig. 3c in which the Pb, Ti, and Sr atoms are denoted by blue, yellow, and red balls, respectively. The shift of the Ti atoms to the right indicates the dipole points to the right, indicating the polarization direction is to the right. Consequently, the STO nanolayers selectively grown on the positive polarization facet led to the exposure of the negative polarization facet[46].

To investigate the impact of the STO nanolayers on the direction of charge separation, we performed reduction photodeposition of Ag, oxidation photodeposition of $MnO_x$ (Mn in a valence state higher than divalent), and co-deposition of Ag and $MnO_x$ on PTO-STO, as shown in Fig. 3d–f. The deposition processes could be expressed as the following path:

$$Ag^+ + e^- \rightarrow Ag$$

$$Mn^{2+} + (2x - 2)h^+ \rightarrow MnO_x (x > 1)$$

during the photodeposition process, $Ag^+$ gained electrons and formed Ag, while $Mn^{2+}$ gained holes and formed a higher valence state $MnO_x$. The deposition of Ag on the STO film (Fig. 3d) indicated that electrons

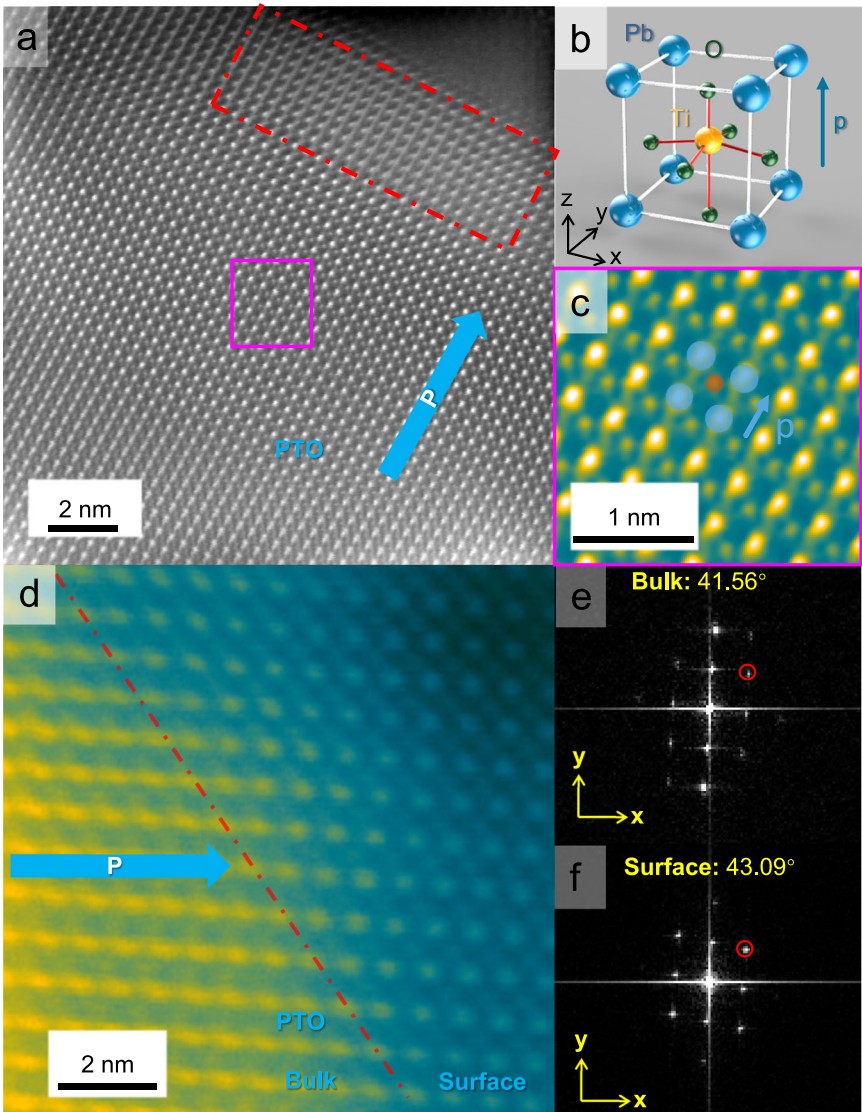

**Fig. 2 | Surface structure of PTO. a, d** STEM of PTO near the positive polarization surface, the blue and yellow dot lines are the atom row of PTO in bulk and surface, respectively. The red dotted line refers to the bulk and the surface bound. The blue arrow points to the polarization direction. **b** Schematic of the unit structure in the PTO cell, the blue, yellow, and green balls represent Pb, Ti, and O atoms, respectively. The black arrows labeled 'x, y, z' show the coordinates axis of the unit cell. The blue arrow points to the direction of polar in the PTO unit cell, which is the same as the shift direction of the Ti atom in the unit cell. **c** magnified HADDF STEM of PTO in the magenta region of (**a**). Blue and orange balls represent Pb and Ti atoms, respectively. The blue arrow points in the direction of polar. **e, f** The FFT results of bulk and surface region, respectively. The yellow arrows labeled 'x, y' defines the coordinates axes in the FFT images. The circle represents the (101) direction of PTO, and the angle is the angle between the (101) point and the x-axis.

were separated to STO film, while the deposition of MnO$_x$ on uncovered PTO (Fig. 3e) showed holes being transferred to the exposed PTO surface. Furthermore, the co-deposition results indicated electrons and holes were transferred in different directions (Fig. 3f). Meanwhile, surface photovoltage (SPV) tests were conducted to confirm the charge separation direction. It can be seen from Fig. 3g that the negative polarization facet without STO coverage shows a positive SPV response of hole transfer to the surface under 400 nm illumination (details in Supplementary Fig. S9). Conversely, the positive polarization facet covered by STO nanolayers (Fig. 3h) shows a negative SPV, signifying the transfer of electrons to the STO surface (details in Supplementary Fig. S10). Importantly, the introduction of STO nanolayers does not alter the charge separation direction in PTO, i.e., holes transfer to the negative polarization facet while electrons are transferred to the positive polarization direction.

Since the presence of the defective structure near the surface of bare PTO has been confirmed in Fig. 2, the nature of the defects should

be further elucidated. Figure 4a, b presents the results of time-of-flight secondary ion mass spectrometry (TOF-SIMS). In Fig. 4a, the TOF-SIMS results for PTO samples revealed a gradual decrease in Pb content from the surface to the bulk. In contrast, the Ti content exhibited a significant increase from the surface to the bulk. The TOF-SIMS results for PTO-STO samples (Fig. 4b) revealed that, in contrast to PTO samples, the Ti content remains constant from the interface to the bulk of PTO (details in Supplementary Figs. S11 and S12, and the calibration in Supplementary Fig. S13). In conjunction with the STEM results, these findings provide a clear picture that the near-surface region of PTO undergoes distortion due to the leaching of Ti atoms. The subsequent growth of STO nanolayers successfully eliminates these Ti defects, leading to the restoration of the lattice structure. In addition, the reduced distortion regions in the STO surface indicate fewer defect regions on the STO surface (Supplementary Fig. S14).

The evolution of the structure from PTO to STO was illustrated in Supplementary Fig. S15. After the growth of STO nanolayers, the

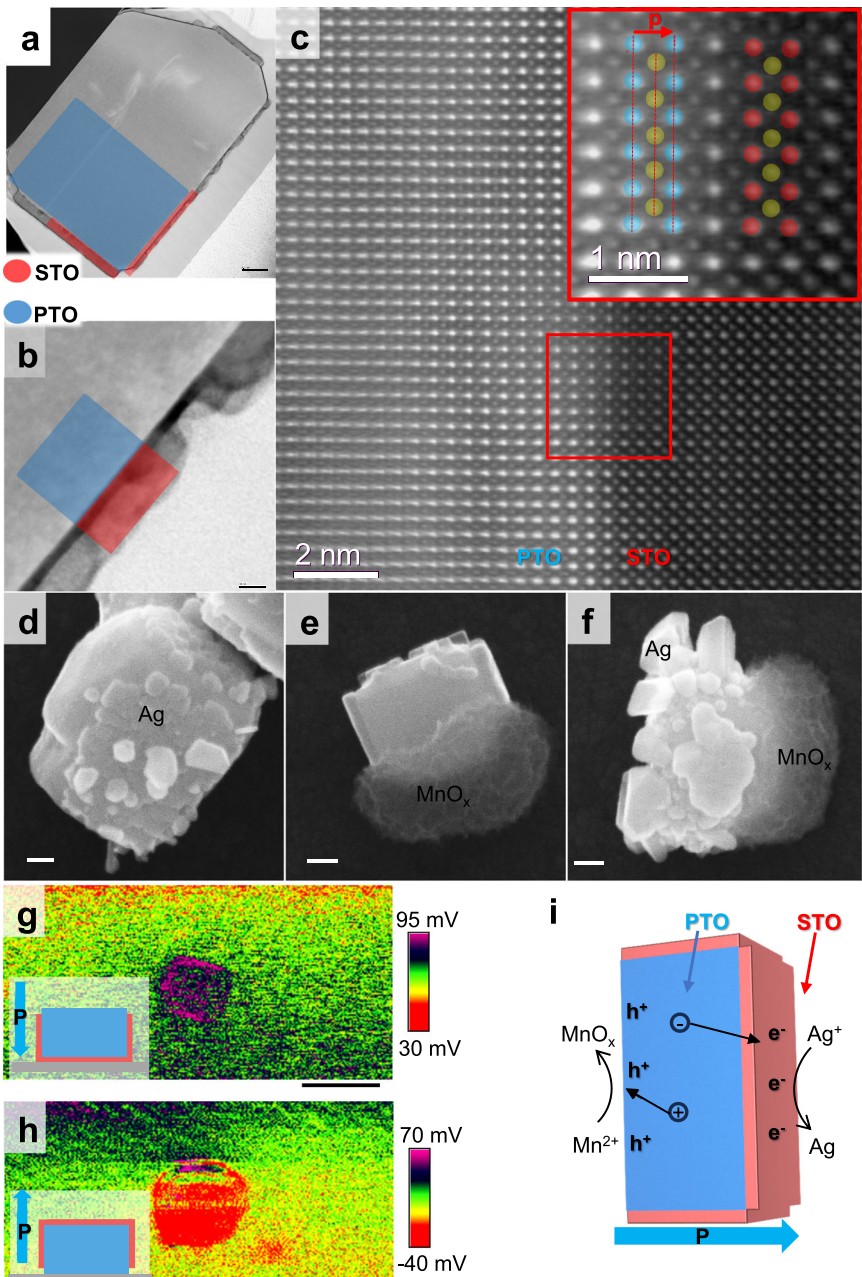

**Fig. 3 | Selective growth of STO on PTO and the charge separation image studies. a, b** SEM of PTO-STO structure. **c** STEM of the interface between PTO and STO on the positive polarization facet. The picture at the top right is magnified from the red rectangle, where the blue, yellow, and red balls represent Pb, Ti, and Sr atoms, respectively. The red arrow labeled 'p' points to the direction of polar in the PTO unit cell. **d, e** photodeposition results of Ag and $MnO_x$ on PTO-STO particles, respectively. **f** photodeposition of both Ag and $MnO_x$ on PTO-STO particles. **g, h** SPVM of PTO-STO samples with polarization direction down and up, respectively. The inset shows the exposed surface and the polarization direction where the blue region was PTO, the red region was STO, and the depolarization direction was indicated by the blue arrow. **i** the schematic of the charge separation direction and photodeposition process in the PTO-STO structure. The black arrows show the direction of electrons or hole migration. The blue and red arrows point to the region of PTO and STO in the schematic, respectively (The scale bars: (**a, b, d, e, f**) 100 nm, (**g, h**) 800 nm, respectively).

electronic structure in the bulk region of the sample remained similar to that of bare PTO, with distinct splitting between the $t_{2g}$ and $e_g$ peaks observed. In contrast, the splitting between the $t_{2g}$ and $e_g$ peaks was larger in bulk STO due to its cubic phase and higher symmetry. Interestingly, at the PTO surface where STO nanolayers were grown, the splitting between $t_{2g}$ and $e_g$ peaks resembled that observed in bulk STO, suggesting that the PTO surface region acquired some cubic-like characteristics. Moreover, the transition between $t_{2g}$ and $e_g$ peaks splitting from bulk to surface regions exhibited an opposite trend

compared to bare PTO, as illustrated in Supplementary Fig. S4 and previously discussed. This cubic-like structural feature was further confirmed by the repair of Ti defects and the substitution of Pb with Sr during STO growth, as seen in Supplementary Fig. S16. For a clearer comparison of the structural changes at the PTO surface before and after the growth of the STO nanolayer, we extracted and analyzed EELS data from the PTO surface before and after STO growth. Figure 4c shows that the surface structure of PTO, initially characterized by defective regions with the minimal splitting of the $t_{2g}$ and $e_g$ peaks,

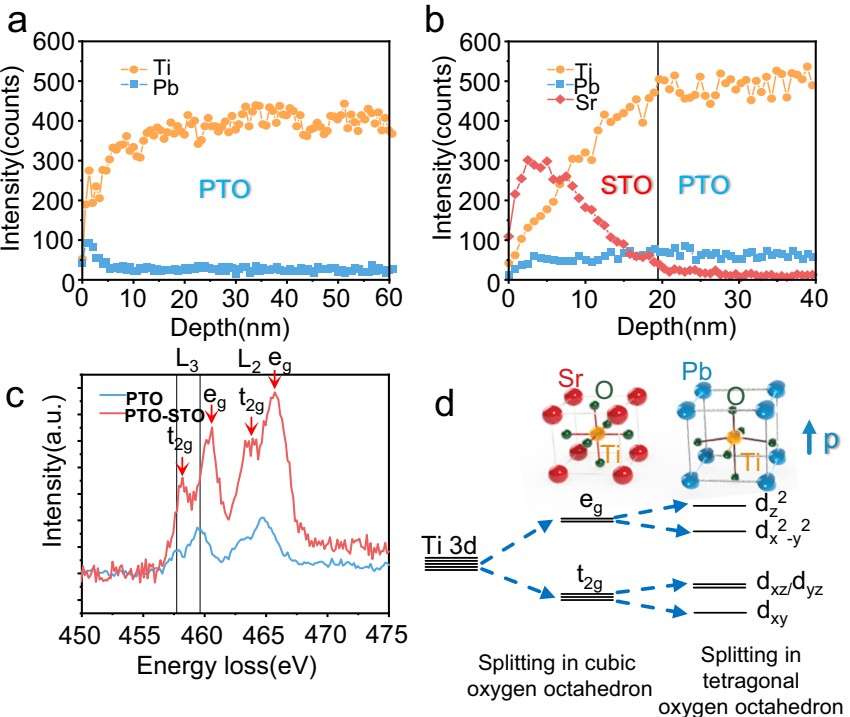

**Fig. 4 | Surface structure changes after STO growth. a, b** Element distribution profile across the positive polarization direction before and after the growth of STO nanolayers as measured by TOF-SIMS, respectively. **c** EELS data of the PTO surface before and after the growth of STO nanolayers. The black vertical line marks the position of L$_3$ peaks of the PTO surface. The red arrows point to the peaks of Ti t$_{2g}$ and e$_g$. **d** the schematic of splitting of d orbitals of Ti in oxygen octahedron field. The blue arrow labeled 'p' points to the direction of polar in the PTO unit cell. The black lines represent the d orbits of Ti. The red, blue, yellow, and green balls represent Sr, Pb, Ti, and O, respectively.

transforms to exhibit more pronounced peak splitting after STO modification. These changes indicated a reduction in defect density and a transition to a more ordered interface between PTO and STO.

The peak splitting could be explained by the schematic in Fig. 4d. In the cubic phase of STO, the Ti atom is located at the center of the oxygen octahedron. In this configuration, the Ti d-orbitals split into t$_{2g}$ (d$_{xy}$, d$_{xz}$, d$_{yz}$) and e$_g$ (d$_{x^2-y^2}$, d$_{z^2}$) levels. In contrast, in the tetragonal phase of PTO, the elongation along the z-axis and contraction in the x- and y-axes lead to a reduction in symmetry. This, in turn, causes Ti to be displaced from the center of the oxygen octahedron, further lifting the degeneracy along the x, y, and z-directions. Consequently, the t$_{2g}$ and e$_g$ orbitals performed further splitting in each peak and experienced less splitting between t$_{2g}$ and e$_g$ peaks. Meanwhile, at defective regions of PTO surfaces, further distortion may exacerbate the reduction in degeneracy along the x and y directions, resulting in increased splitting within t$_{2g}$ orbitals and a reduction in the splitting between t$_{2g}$ and e$_g$ peaks. This observation explains the differences in peak splitting between defective PTO and STO surfaces and provided evidence that surface defects were eliminated with the growth of STO nanolayers.

To figure out whether the elimination of surface defects could facilitate the overall water splitting, we conducted overall water splitting reactions. The bare PTO yields H$_2$ at 3.27 µmol/h, after the introduction of STO nanolayers, the rate substantially increased to 216.83 µmol/h. Surprisingly, the STO/PTO shows a nearly 65-fold enhancement in photocatalytic overall water splitting activity (Fig. 5a, b). The apparent quantum yield (AQY) reaches 4.08%@365 nm, which is the highest efficiency reported so far for ferroelectric photocatalysts. The more remarkable enhancement of activity in our samples compared with recent similar PTO-STO structures stems from the different defect structures between the samples. More defects in pristine PTO result in lower activity. In contrast, the reduction of

defects after modifying PTO with STO nanolayers leads to higher activity. This difference arises from variations in the preparation conditions (see Supplementary Fig. S17).

To exclude the specific contribution of STO nanolayers to the photocatalytic water-splitting reaction, a control experiment was conducted wherein PTO was mixed with STO in a 10:1 molar ratio (equivalent to the ratio of the raw materials during the growth of STO nanolayers). The resulting H$_2$ production rate was approximately 20 µmol/h, and the AQY was 0.2%@365 nm, which is significantly lower than that of PTO-STO samples. This discrepancy suggests that the boosted reactivity is due to the synergistic effect between PTO and STO.

Although enhanced photocatalytic activity was observed following the growth of STO nanolayers on PTO, the precise mechanism underlying this improvement remains elusive. Previous findings suggested that the deposition of STO nanolayers not only reduced surface defects in PTO but also promoted the formation of a transition interface between the two materials. Building on these observations, we proposed two potential roles for the STO nanolayers in enhancing photocatalytic activity: The STO nanolayers could form a heterojunction with PTO. This heterojunction might significantly contribute to the observed enhancement in photocatalytic activity by facilitating charge separation and transfer at the interface. In this scenario, charge generation would occur within the space-charge region (SCR) at the PTO/STO interface, where the depletion layer aids in separating electrons and holes under illumination. It is worth noting that, unlike conventional centrosymmetric semiconductors, the depolarization field in PTO can further assist in electron and hole separation; Alternatively, the STO nanolayers might primarily serve as a surface modification layer, altering the surface structure and properties of PTO. In this case, the charges involved in water splitting would be predominantly generated within the PTO particle itself. The STO layer

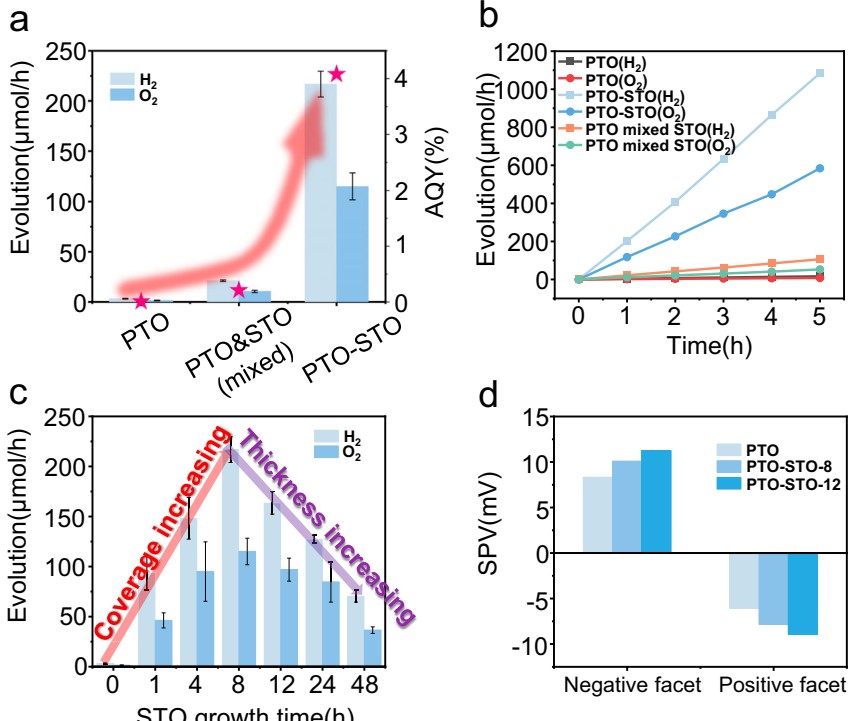

**Fig. 5 | Overall water splitting activities with the modification of STO nano-layers. a** overall water splitting activities and AQY (shown by pink stars) of PTO, PTO mixed with STO and PTO-STO samples, the red arrow showed the change in water splitting activities in different samples. The error bar represents the standard deviation. **b** $H_2$ and $O_2$ evolution of PTO, PTO mixed with STO and PTO-STO samples.

**c** Overall water splitting activities of PTO and PTO modified by STO nanolayers with different growth times. The red arrow shows the increased activities with the increased STO coverage, and the purple arrow shows decreased activities with the increased STO thickness. The error bar represents the standard deviation. **d** SPV signals of PTO and PTO modified by STO nanolayers with different growth time.

would then enhance the surface charge dynamics within PTO, possibly by improving charge transport or reducing charge recombination. Charge separation in this scenario would primarily originate within PTO, driven by its ferroelectric depolarization field. To ascertain which of these mechanisms plays a dominant role in the enhanced photocatalytic activity observed, further investigation is required.

To better understand the function of STO, we prepared samples with the growth of STO nanolayers on PTO for different growth times. These samples are denoted as PTO-STO-x, where 'x' represents the duration of STO growth on PTO. We then investigated their photocatalytic activities and SPV signals to assess their performance. As depicted in Supplementary Figs. S18 and S19, the STO nanolayers' growth process on PTO surfaces can be divided into two distinct stages. During the initial stage, driven by the electrostatic interactions, the STO precursor adsorbed onto both the positive depolarization and nonpolar surfaces of PTO, and gradually, a thin film would form within 1 to 8 h. After reaching the 8 h mark, nearly complete and uniform film coverage was achieved. Further prolonging the reaction time initiated the second stage, characterized by a notable increase in film thickness. These two stages also correspond to changes observed in the overall water-splitting reaction, with the amount of STO loading increasing over time.

Figure 5c provides a clear visualization: as the coverage of STO nanolayers on PTO increased, the reactivity exhibited an upward trajectory, reaching its peak when a fully covered film was formed at the 8 h mark. However, any further increase in STO nanolayers' thickness beyond this point results in a decrease in reactivity (details in Supplementary Fig. S20). Conventionally, an increased width of the SCR is expected to enhance charge separation and consequently, reactivity. However, the reactivity exhibits a volcano-shaped pattern, suggesting a more complex relationship between SCR width and reactivity. It should be noted that the maximum thickness of STO nanolayers on

PTO-STO is approximately 65 nm, which is significantly smaller than the typical width of the SCR in heterojunctions (typically 150–200 nm with slight doping). In addition, it has been observed that some Ti defects also exist in the STO layer. As the thickness of the STO layer on the PTO surface increases, electrons would diffuse over a greater distance within the STO film. This extended diffusion length would increase the charge recombination probability through defects in the STO layer, which in turn leads to decreased activities for water splitting. As shown in Fig. 5d, the SPV results of PTO samples modified with STO nanolayers show only a slight increase as the growth time increased, which can be attributed to the increased SCR width. Interestingly, PTO-STO-12 samples, despite having the highest SPV values, exhibit decreased reactivity in photocatalytic water splitting. These findings suggest that the slightly enhanced charge separation ability may not be the main contributor to the observed increase in activity by two orders of magnitude. In addition, after annealing above the Curie temperature, the activity was reduced by nearly half, which indicated the majority of carriers came from PTO rather than STO, and that charge separation was primarily driven by the ferroelectric depolarization field rather than the heterojunction structure. These results indicate that STO nanolayers growing on PTO may have other influences on water splitting, rather than forming the heterojunctions. From the previous SEM images of the PTO-STO structure in Fig. 3b and Supplementary Fig. S22, the STO on the positive polarization direction of PTO exhibited a particle morphology, with gaps emerging between the STO particles. To determine whether the electrons separated to STO contributed to water splitting, the position of the hydrogen evolution cocatalyst, Rh, was investigated. Infrared (IR) spectroscopy results (Supplementary Fig. S23) confirmed that Rh was located on the STO film rather than in the gaps between the STO particles and PTO.

To further investigate the specific role of STO nanolayers and the elimination of Ti defects to carrier dynamics, we performed transient

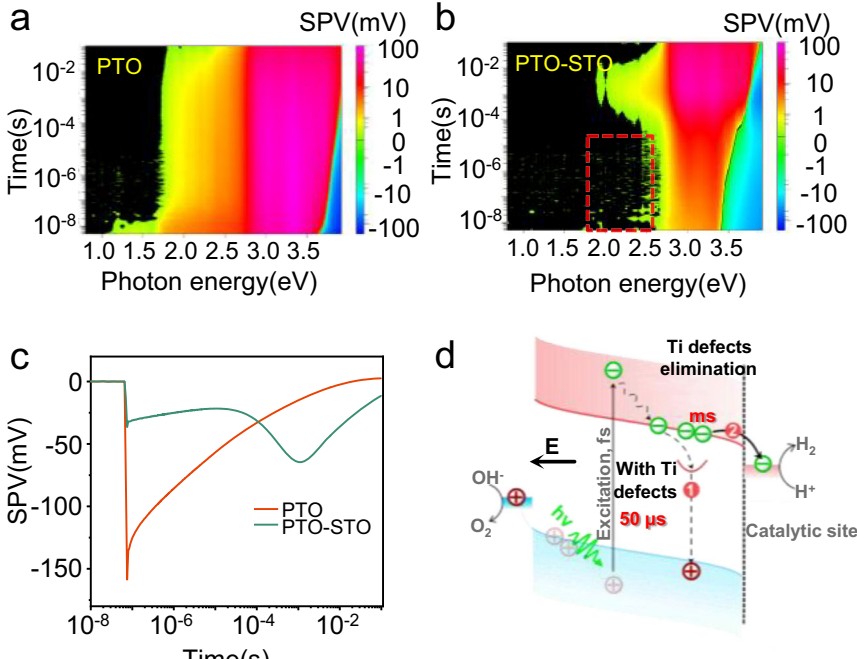

**Fig. 6 | Charge transfer dynamics in PTO and PTO-STO. a, b** TPV data of PTO and PTO-STO samples, respectively. **c** Transient SPV data of PTO and PTO-STO samples under 3.95 eV excitation. **d** the surface charge dynamic of electrons in PTO (marked by process 1) and STO-modified PTO (marked by process 2) samples. The black arrow labeled '**E**' points to the electric field direction in PTO. The red balls labeled '+' indicate holes, and the green balls labeled '−' indicate electrons. The red arc indicates Ti defects-related states. The black, short, dotted line indicates the surface of PTO. The blue regions and pink regions in the middle represent the valence band and conduction band, respectively. The blue regions in the left region are oxidation sites, and the pink regions in the right region are the reduction sites.

surface photovoltage (TPV) experiments. The TPV pseudo-color spectra of PTO and PTO-STO are presented in Fig. 6a, b. Both spectra show a significant SPV signal upon superband excitation with photon energy higher than 2.95 eV. A comparison of the two spectra before and after STO modification shows the disappearance of the SPV signals within a time frame of less than $10^{-5}$ s upon the modification of PTO with STO nanolayers. This is evidenced by the red rectangle region shown in Fig. 6b, which was measured by exposing it to light energy ranging from 1.7 to 2.6 eV. Typically, SPV signals emerging under sub-bandgap light illumination are associated with defects. Accordingly, the different signal observed in PTO-STO samples is an indication of the elimination of the Ti defects in the surface region of the PTO surface after the growth of STO nanolayers. In addition, the signal from STO can be disregarded due to its much weaker intensity compared to that of the PTO samples, as shown in Supplementary Fig. S24.

To investigate the consequence of fewer surface defects and a well-defined interface structure on electron dynamics, we conducted a middle infrared transient absorption spectroscopy (MIR-TAS) (Supplementary Fig. S25a), and the corresponding fitted data are provided in Supplementary Table S1. The 5000 nm probe MIR light is used to monitor the free electrons in the conduction band and shallow trapped electrons below the conduction band[47]. Both STO and PTO samples exhibit typical absorption signals. The dynamic fit of bare PTO unveils three-lifetime profiles for electron decay: $22.4 \pm 2$ ps, $246 \pm 13$ ps, and $3000 \pm 800$ ps, respectively. Interestingly, the PTO-STO samples show a reverse and long-lived bleach signal. The bleach signal was likely related to some interface states (Supplementary Fig. S25). Notably, these interface states had little influence on charge transfer dynamics, as will be demonstrated in the subsequent data in Fig. 6c. The fitted lifetimes of the electrons in PTO-STO samples are $14.8 \pm 2.1$ ps, $520 \pm 40$ ps, and $10000 \pm 2600$ ps, respectively. The significantly increased lifetime from 3000 ps to around 10000 ps is consistent with the reduction of surface defect density. The slow transient absorption

spectroscopy further confirms the retardation of recombination by the growth of STO nanolayers (Supplementary Fig. S26).

To further explore surface charge dynamics, transient surface photovoltage (TPV) studies were conducted. As illustrated in Fig. 6c, both PTO and PTO-STO samples exhibit negative signals, indicative of electron transfer to the surface. The fitted results in Supplementary Table S5 reveal a two-step electron recombination process in PTO. One process is associated with quick recombination through shallow trap states, while the other is due to recombination through deep trap states[48]. The dominant process is recombination via the deep trap state, as evidenced by the much larger fitted amplitude value of −121 mV compared to that at the shallow state ($-2 \times 10^{-4}$ mV). The fitted lifetime indicates that electrons are captured and recombined at 50 µs (see Supplementary Table S5).

After the growth of STO nanolayers, the fitted results exhibit three processes. The first is weak recombination (amplitude $-3 \times 10^{-5}$ mV) through shallow traps at a timescale shorter than $10^{-5}$ s. The increasing signal from $10^{-4}$ to $10^{-3}$ s might be attributed to the diffusion of electrons in the non-ferroelectric STO[49]. The subsequent signal from $10^{-3}$ to $10^{-1}$ s represents electron recombination at the surface deep trap state, with a fitted lifetime of 9 ms, indicating a significantly extended lifetime of the electrons compared with that of pure PTO (50 µs). The extended lifetime of electrons in PTO-STO samples indicates a significant reduction in recombination, which can be attributed to the elimination of surface defects. Meanwhile, the decreased photoluminescence (PL) peak after the growth of STO nanolayers indicated reduced recombination, likely due to the elimination of surface Ti defects in PTO.

It is therefore concluded that the capture and recombination processes of these defects on the surface of bare PTO samples hinder electrons from further participating in water splitting (Fig. 6d). The decrease in defects may result in the significant elimination of these undesired processes, thereby improve the photocatalytic water splitting activity.

In summary, this study unveils the relationship between surface charge dynamics and surface reactions in the ferroelectric photocatalyst of $PbTiO_3$. Initially, the existence of Ti defective structure on the surface of $PbTiO_3$ was confirmed through high-resolution imaging and electron energy loss spectroscopy. These Ti defects can be largely eliminated by the growth of $SrTiO_3$ nanolayers on the surface of $PbTiO_3$, resulting in a well-defined interface structure of $PbTiO_3$, as evidenced by high-resolution imaging and secondary ion mass spectrometry.

The defects-reduced structure resulted in the development of a highly efficient ferroelectric photocatalyst for overall water splitting producing $H_2$ 216.83 at µmol/h and $O_2$ 116.9 at µmol/h, achieving a remarkable apparent quantum yield of 4.08%. Our investigation also reveals the significant impact of Ti defect elimination on carrier dynamics near the positive interface. Specifically, the electrons separated to the positive polarization facet could be trapped by the defective structures, impeding their participation in water splitting reaction. The introduction of $SrTiO_3$ nanolayers largely eliminates Ti defects, successfully prohibiting the trapping process and leading to a prolonged electron lifetime from 50 µs to ms scale. Moreover, the observation of a bleach signal after the growth of $SrTiO_3$ nanolayers confirms the repair of the surface structure of the ferroelectric material.

Overall, our findings offer a comprehensive understanding of the influences of surface defects on ferroelectric charge separation and transfer to the surface catalytic sites. This study underscores the effectiveness of eliminating defective structures through chemical growth of nanolayer methods in substantially boosting of photocatalytic activity. The clarity of this defect elimination approach holds significant potential for further enhancing photocatalytic performance.

# Methods
## Preparation of single domain $PbTiO_3$ crystal
$PbTiO_3$ single domain crystals were prepared by hydrothermal method. 0.2396 g Anatase $TiO_2$(Alfa, 99.6%), 1.2421 g $Pb(NO_3)_2$(Aladdin, 99%) and 3.36 g KOH(Aladdin, 95%) were added in 5 mL, 10 mL and 15 mL deionized water with resistance of 12 MΩ, respectively. Then $Pb(NO_3)_2$ solution was dropped to KOH solution. After a 5 min stirring, $TiO_2$ mixture was added in the mixture solutions and stirs for 5 min. The resulting solutions were then transferred to a 50 mL Teflon-lined reactor and heated at 260 °C for 36 h. After the reaction, the yellow powders were washed 3–5 times with deionized water and dried at 80 °C.

## Preparation of single $PbTiO_3$-$SrTiO_3$ composite
2.4 g NaOH(Aladdin, 97%) was dissolved in 15 mL deionized water with resistance of 12 MΩ, and then 3 mL 0.1 mol/L tetrabutyltitanate (Sinopharm Chemical Reagent, 98%)(dissolved in 1 mol/L $HNO_3$(Sinopharm Chemical Reagent, 65 - 68 %)) was added in NaOH solution with stirring. After that, 12 mL deionized water with resistance of 12 MΩ containing 0.0635 g $Sr(NO_3)_2$(Sinopharm Chemical Reagent, 99.5%) was added to the mixture. Finally, 0.909 g prepared $PbTiO_3$ was added. After a 30 min stirring, the mixture was transferred to a 50 mL Teflon-lined reactor and heated at 200 °C for varying reaction times. After the reaction, the resulting products were washed 3–5 times with a combination of ethanol and water and subsequently dried at 80 °C.

## Cocatalyst loading
Cocatalyst loading was achieved by photodeposition during the reaction. 200 mg of PTO or PTO-STO powders were dispersed to the same vessel containing 100 mL distilled water as used for the photocatalysis reaction. After brief sonication, 100 µL of 2 mg/mL $RhCl_3\cdot3H_2O$ (Alfa, 99.95%) solution was added to the vessel under magnet stirring and irradiation by 300 W Xe lamp for 10 min. Subsequently, 50 µL of 2 mg/mL $K_2CrO_4$ (Sinopharm Chemical Reagent, 99.5%) solution was introduced to the suspension and irradiated for 5 min. After a brief time of evacuation, 50 µL of 2 mg/mL $Co(NO_3)_2\cdot6H_2O$ (98.5%) solution was added and irradiated for 5 min. As results, Rh, $CrO_x$, and CoOOH could be deposited.

## Photocatalysis test
The photocatalytic reaction was conducted by placing the high borosilicate glass reaction vessel, with a quartz cover and containing 100 mL of deionized water, in a water bath at 15 °C. The reaction was in a closed-loop reactor and irradiated with a 300 W Xe lamp. A total of 200 mg of the photocatalyst, with cocatalyst loaded in situ without further purification, was used. Sampling for product analysis via GC was performed at one-hour intervals. After each sampling, the reaction vessel was subjected to vacuum for 10–15 min. The entire reaction process was carried out using the Perfectlight Sci&Tech reactor. For quantifying $H_2$ and $O_2$ production, external standard quantification was employed. Different amounts of $H_2$ and $O_2$, in a 2:1 ratio, were successively injected into the reactor containing 100 mL of water. After allowing 30–60 minutes for equilibrium, the gas was injected into the GC with an Ar carrier. A series of data were acquired, and the plot of the injected amounts of $H_2(O_2)$ versus the peak areas provided the reactor's calibration coefficient. During the test, the amounts of $H_2$ and $O_2$ production were calculated using this coefficient and the peak areas of $H_2$ and $O_2$.

## Kelvin probe force microscopy (KPFM)
The PTO particles were dissolved in distilled water and subsequently deposited onto a conductive tin oxide (ITO) substrate. Following the evaporation of water, the samples were ready for KPFM testing. The contact potential difference (CPD) was measured using the amplitude-modulated KPFM module on a Bruker Dimension Icon microscope. The CPD value was determined by measuring the potential difference between the sample's work function and the SCM-PIT-V2 probe, characterized by a tip radius of ~25 nm. To maintain a reliable signal-to-noise ratio, the probe was lifted by 50 nm after each morphology scan. To acquire CPD measurements under illumination, samples were exposed to a monochromatic light with energy levels ranging from approximately 0 to 2 mW, the light source was an Xe lamp passed through a grating before reaching the samples.

## Surface potential voltage spectrum (SPS)
The SPS test was performed using a home-modified KFPM module. Following the initial KPFM imaging test, the probe was positioned at the specific point of interest, and a beam of monochromatic light modulated by a chopper was directed onto the testing area. The amplitude (R) and phase (θ) signals were acquired using a Stanford SR830 lock-in amplifier by comparing the variations in surface potential with the modulated signals. The magnitude and direction of charge separation were determined from the amplitude and phase data, respectively. This experimental setup allowed for precise quantification of charge distribution dynamics during the SPS test.

## Transient surface photovoltage spectroscopy
Transient surface photovoltage spectroscopy measurements were conducted using a fixed-capacitor arrangement. The SPV signals were coupled with a high-impedance buffer (measurement resistance 50 G). Modulated illumination was provided by a halogen lamp with a quartz prism monochromator (SPM2). SPV transients were excited using pulses from a tunable Nd:YAG laser (EKSPLA NT342/1/UVE) with a pulse duration of 4 ns and a repetition rate of 1 Hz, respectively. The transients were detected with a resolution time of 5 ns (GaGe CS 14200). A logarithmic read-out with a logarithmic averaging procedure was applied for the registration of SPV transients.

## Piezoresponse force microscopy (PFM)
The preparation of PFM samples was carried out using the following method: First, the particles were dispersed on a monocrystal polished

Si surface. Next, a layer of Au film was evaporated onto the Si surface. Subsequently, the Au film was adhered to a conductive ITO surface using an adhesive layer. Finally, the Si substrate was peeled away, transferring the particles and Au film onto the ITO substrate. Before testing, electrical contact was established by connecting the Au film and exposed ITO with conductive Ag adhesive The PFM experiments were conducted using a Bruker Dimension Icon microscope equipped with a SCM-PIT-V2 conductive probe. To enhance imaging quality, the contact resonance mode was employed. PFM hysteresis loops were measured under the same conditions as the imaging process.

## Mid-infrared transient absorption spectroscopy (MIR-TAS)

The mid-infrared femtosecond transient absorption spectra were recorded using a custom-built experimental setup, identical to the one described in our previously reported article[50]. Briefly, a femtosecond titanium sapphire laser (800 nm, 35 fs, 1 kHz, Spitfire Ace, spectra Physics) served as the laser source. The excitation beam was generated by an optical parametric amplifier (TOPAS, Spectra-Physics), covering a spectral range from ultraviolet (350 nm) to mid-infrared (IR) (6 μm). During the test, the excitation beam was turned to 360 nm with an energy of 60 μJ/cm$^2$. The supercontinuum mid-IR probe pulse covering a wavelength range of 2.5 – 10 μm, was obtained by four-wave mixing, where the 800 nm laser was frequency doubled to 400 nm, and then the resulting light was focused in air to produce the mid-IR supercontinuum probe pulse. During the test, the probe beam was turned to 5 μm. The mid-IR probe light was detected using a liquid-nitrogen-cooled 64-element mercury-cadmium-telluride-coupled spectrometer (FPAS-0144, Infrared Systems Development Corporation).

## Slow transient absorption spectroscopy (TAS)

TAS measurements were performed using the same method as described in our previously reported study[51], employing the third harmonic of a Nd:YAG laser (Opolette HE-355-LD, 355 nm, 7 ns pulse width, 205 μJ/pulse, 1.05 s period) as the pump source. A liquid light guide transmitted the laser pulse to the sample resulting in an incident pump intensity of ca. 205 μJ/cm$^2$. A 100 W tungsten lamp (Bentham, IL 1) coupled to a monochromator (Zolix, Omni - λ 300) was used as the probe light (900 nm). Variation in optical density (ΔOD) of the sample was calculated by measuring the transmitted light using a Si photodiode (Hamamatsu) and an amplification system coupled to both an oscilloscope (Tektronix, TDS 2012C) and data acquisition card (National Instruments NI-6221). The data were averaged over 300 laser shots. The samples were prepared by tableting a mixture of PTO or PTO-STO and KrBr (Aladdin) with a mass ratio of 1:10 under a pressure of 8 MPa.

## Diffuse reflectance infrared Fourier transform spectroscopy (DRIFTS)

To confirm the position of Rh on PTO-STO samples, DRIFTS measurements were conducted. Before each measurement, the sample was reduced by $H_2$ at 300 °C for 30 min with a flow rate of 20 mL/min and purged with $N_2$ within 20 mL/min at 300 °C for 30 min. Subsequently, the sample was cooled to 30 °C in Ar atmosphere. Afterward, CO gas was injected into the cavity, followed by purging with Ar to remove the CO from the cavity. Finally, IR data were acquired.

## Structure characterization

SEM morphology tests and EDS element imaging were performed on Quanta 200 F and JSM-7900F. High-resolution STEM test and EDS element imaging were obtained using JEM-300F. the channel was set to 0.15 eV/channel, and the pixel size was 0.5 nm × 0.5 nm × 0.15 eV the pixels during PTO samples, the pixel dimensions were 195 × 215, and for the PTO-STO samples, the pixel dimensions were 62 × 162. The STEM and EELS samples were prepared using a physical slicing method with a diamond blade. Specifically, the particles were embedded in resin, and after the resin hardened, the block was cut into

approximately 150 × 200 nm pieces with a diamond blade. These pieces were then sliced into approximately 50 nm sections, which were transferred to a copper mesh for testing. This method minimizes damage from the ion beam.

## Materials characterization

XRD patterns of samples were collected on a Rigaku D/max 2500 spectrometer using Cu Kα X-ray as the radiation source. SIMS results were obtained from M6 TOF-SIMS. EPR data were obtained from Bruker A200 at room temperature.

## Data availability

All data generated and analyzed in this study are included in the article and Supplementary Information and are also available at the corresponding authors' request. Source data are provided in this paper.

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

## Acknowledgements

This work was conducted by the National Program on Key Basic Research Project (2021YFA1500600, 2023YFA1507003), the Funda-mental Research Center of Artificial Photosynthesis (FReCAP), National Natural Science Foundation of China (22325205, 22088102), CAS Pro-jects for Young Scientists in Basic Research (YSBR-004), the Strategic Priority Research Program of the Chinese Academy of Sciences (Grant No. XDB0600300), Fundamental Research Funds for the Central Uni-versities, 20720220011. This work has been supported by the New Cornerstone Science Foundation through the XPLORER PRIZE. We thank Fusheng Li and Chang Liu from Dalian University of Technology for their assistance with the tests.

## Author contributions

J.Z. conceived and conducted most of the experiments and analyzed data; Yong Liu offered some help in sample preparation; T.D. conducted transient TPV experiments and analyzed data; Y.W. helped analysis MIR-TAS data and make revisions in the MIR-TAS part; Z.W. help analyzed MIR data; Y.X. conducted MIR-TAS experiments and analyzed data; G.L. supported STEM sample preparation; C.Z. helped prepare STEM sam-ples. D.S. helped analyze STEM and EELS results. P.J. conducted STEM and EELS experiments and analyzed data; M.L. conducted STEM and EELS experiments; N.T. conducted STEM and EDS experiments; H.Z. conducted SIMS experiments; D.L. conducted slow-TAS experiments and analyzed data. X.W. conducted slow-TAS experiments and analyzed slow-TAS and MIR-TAS data; Z.F. conducted DRIFTS experiments and analyzed data; Z.L. supported in photocatalysis experiments; Yaling Luo helped in photocatalysis experiments and AQY test; J.C. helped picture design; F.F. conceived most of the experiments and analyzed data. The manuscript was written by J.Z. and F.F.; C.L. proposed the project, analyzed data, and revised the manuscript.

## Competing interests

The authors declare no competing interest.
