## [Peer Review File · Nature Communications]

Unveiling Charge Utilization Mechanisms in Ferroelectric for Water Splitting

Corresponding Author: Professor Can Li

Version 0:

Reviewer comments:

Reviewer #1

(Remarks to the Author)

The authors use here a large suit of characterization techniques to probe the mechanisms why photocatalytic activity of the PTO improves when coated with thin layers of STO. The first part of the manuscript is well written. The results and discussion part is somewhat cluttered due to use of many techniques and the results being shown in short snippets to justify the hypothesis of the authors. However, I have multiple reservations on the data presented hence find the validity of the presented results questionable in the presented format.

1. Figure 1d (bulk region) and Figure 4b P1 (in the bulk region) does not seem to agree and e_g feature is missing in P1 region.
2. Negative control group reference measurement of 10:1 PTO and STO mixture shows a much stronger 20 $\mu\text{mol/h}$ H₂ production rate vs. only PTO showing 3.27 $\mu\text{mol/h}$. This represents a 6-fold improvement without any treatment. Thus, bare PTO may not a realistic comparison point, hence 400-fold improvement may be an over estimation.
3. Figure 4 is misleading. X-axis scale in Figure 4c is very narrow whereas in Figure 4d it is much wider range. Also, y-axis is not specified in values. Thus, Figure 4c vs. Figure 4d comparison is currently not meaningful.
4. Some of the authors from this work have studied the same PTO/STO system for its photocatalytic activity Ref. 44. There, they reported maximum H₂ production rate up to 80 $\mu\text{mol/h}$ whereas the obtained value is 200 $\mu\text{mol/h}$ in the current work as a result of STO thickness optimization. It is not clear the hypotheses proposed by Ref. 44 can be refuted by the current work on the mechanism of photocatalytic activity improvements.
5. Overall, the scientific approach of the authors is rather qualitative. The logical inferences are made based on changes, and sometimes subtle changes, in the experimental data (Ex./ EELS measurements). Thus, the claims of the authors are not clear nor convincing at times.
6. Transient mid-IR probe experiments are somewhat misleading. The assignment of the authors that the signal flip indicates long-lived bleach needs further support by wavelength dependent measurements. The effect may very well arise from some sort of interfacial small energy state. I do not buy the argument on impact ionization. That seems unlikely unless the authors show this carefully with excitation fluence and pump energy dependent studies.

Reviewer #2

(Remarks to the Author)

The article deals with interesting questions regarding the reason behind the low activity of ferroelectric PbTiO₃. It is suggested that low activity is due to the presence of Ti defect at the surface. The authors proposed healing these defects by covering PTO with STO layer. The authors attempt to investigate the interfacial Ti³⁺ defect using EELS. Despite the fact that the analysis shows some differences, the conclusion cannot be strongly supported. In fact, the main results that should confirm the assumption of the 3d splitting into t_{2g} and e_g is not fully convincing. The EELS spectra show that the splitting became lower from bulk to the STO interface, as it was observed for EELS results from the PTO bulk to the PTO surface. In addition, there are many results that are controversial, like the results obtained by TOF-SIMS and EDS. The experimental section is not sufficiently described for the reader, and many information is missing. The results obtained in this article cannot support either the conclusion or the main question raised at the beginning. For all these reasons, I cannot recommend this article for publication.

Bellow more recommendations and comments:

Lack of fair and detailed procedure in the article. At this step, the experimental part cannot be reproduced correctly. Important details in the experimental part are missing:

Page 9: "Fig. 2a and Fig 2b illustrate the cross-section of PTO" How was the sample prepared to check the cross section in STEM?

In the same line, details of the synthesis process and the photocatalytic test are missing. The experimental part should be complemented and detailed:

for example, what was the volume of the water? What is the material of the vessel used? Quartz? sodocalcique glass? Pyrex? What was the volume of the reactor? How was H₂ calculated depending on the dead volume of the reactor? Was it pure water? What was the flux reaching the sample during the photoreaction?

Piezoresponse force microscopy (PFM). What was the adherent used for the PTO to stick in Au film?

NO details are reported in the ms regarding the acquisition of EELS analysis: what are the pixels of the line? Pixel size? Channel? Etc... This step is crucial since it is known that depending on how the samples are prepared, in EELS, the Ti³⁺ signal comes from the damaged foil surfaces.

The degeneracy in the case of Ti⁴⁺ in bulk PTO, which is reported to be partially lifted by the octahedral crystal field, separating 3d into t_{2g} and e_g, e.g., levels, is weak compared to reported studies. No reasonable conclusion can be made from these results since one can argue that it is a simple effect of the Ar ion polishing usually used for the sample preparation for such kind of measurement. The EELS analysis is a bit superficial and should be complemented by an extraction of the Ti³⁺ ratio and compared for different samples using variable preparation.

In addition, it is speculated that STEM analysis shows defect localization. It is surprising how the author comes to such a conclusion.

Once the authors analyzed the separating 3d into t_{2g} and, e.g., the splitting became lower, as was observed by their analysis in Fig. 2 from the bulk to the surface. So why, in one case, is it due to surface defects, and in the other case, it is attributed to defect healing? It is a bit contradictory how it is justified.

In Fig. 2, it is speculated that the positively polarized surface is investigated. However, there is no mention of how positively or negatively charged surfaces are discriminated.

The analysis of the oxidation state of Ti should follow a deep analysis of the ratio of Ti³⁺ from bulk to the interface in PTO and PTO/STO.

Fig. 4, it is speculated that the distortion disappeared. However, more analysis and images of different regions of variable PTO-STO particles should be provided to come to such a conclusion. In fact, the evolution of the EELS spectra from P1 to P5 is very similar to the one recorder in PTO from bulk to the surface, making the conclusion questionable.

The TOF-SIMS profile shows that Ti concentration increases gradually when recording the signal from the surface of STO, and then it becomes constant in PTO. The decrease should be expected in PTO extreme surface and not in the STO. This is surprising and in contradiction with the EELS conclusion, and the EDS analysis. Also, the unit of depth axe is wrong, it should be "nm" (or other unit) and not a.u. The intensity should be "counts".

Figure S4 is neither discussed nor mentioned in the main manuscript.

How apparent quantum yield (AQY) was calculated?

The discussion justifying the decrease of the photoactivity for higher STO thickness is a little weak. The change in the space charge layer at the PTO/STO junction is only related to the potential of the flat bands in each oxide. The increase in the thickness should affect charge diffusion to the surface instead. The discussion should be reviewed in depth and correlated to the reduction of Ti in STO, which could rationalize the decrease in the activity.

The interpretation of the TRABS analysis is quite confusing. The authors attributed the bleaching signal to a shallow trap.

The bleaching is usually attributed to the relaxation of electrons to the ground states. The authors are making conclusions about the presence of shallow traps with no proof. The original spectra of TRABS should shown. If there are any traps, they should appear by new absorption bands.

Reviewer #3

(Remarks to the Author)

This paper investigates the reason for low photocatalytic activity of ferroelectric nanoparticles, relating it to high levels of surface defects, showing that these can be reduced using a surface passivation layer of SrTiO₃. The central question of the paper – why despite their benefits of enhanced charge separation via polarisation, do ferroelectrics still display very low photocatalytic activity – is a very pertinent one in the community. Answering this question will definitely be of benefit to many researchers working on ferroelectric photocatalysis and photocatalytic systems in general. While not giving evidence that this issue is a universal one affecting all photocatalysts, since only a single material is investigated, the evidence that passivation enhances photocatalytic activity in this case is compelling, and will likely prompt researchers to try similar approaches in different systems. The change in photocatalytic activity is backed up by detailed characterisation of the material's structural properties and charge-carrier dynamics, which help to support the conclusions that reduction in structural defects is the key driving factor in the enhanced activity. Certainly, a good correlation is demonstrated between STO coverage/morphology and performance, giving good evidence that it is the surface passivation, rather than STO contribution itself that is driving the enhancement. Finally the researchers report an impressive photocatalytic activity for a ferroelectric catalyst. Therefore, overall I believe the paper is worthy of publication, provided some questions are answered and revisions are addressed.

The most major factor that is not evidenced in the paper, is whether by adding the STO layer the eventual surface defects are almost completely eliminated, or whether they are just shifted to the STO surface, rather than PTO. The TEM work in the paper is excellent, and clearly shows a reduction in structural defects at the PTO/STO interface compared to the bare PTO surface. However, the authors have not included characterisation of the STO surface regions. This raises the question of whether the enhancement in activity is mainly because PTO is much less defected, therefore there is much less recombination in that layer, or also because the catalytic surface presented to the solution (STO in the composite case) is itself much less defected, leading to improved surface catalytic activity. To answer this question, the authors should include detailed TEM characterisation of the STO surface, as well as the PTO/STO interface and PTO surface. The TOF-SIMS in Figure 4d does look like there is some major change in composition around the surface, which suggests some reconstruction/defects. This again supports the need to characterise the STO surface in more detail.

Further, there are a number of minor points that should be addressed before publication:

1. Why are the particles monodomain? For 600 nm particles I would expect them to form multiple domains. The authors should explain/justify this more clearly in the manuscript.
2. Introduction: "memorizer" should either be memristor, or memory.
3. "ferroelectrics are unique in that carriers mostly drift within the depolarization electric field": in most cases depolarisation field is fully screened (otherwise it would depolarise!), so it is not the case that carriers mostly drift with the depolarisation field in ferroelectrics, as this has been screened. Unless the authors mean prior to screening. This point should be clarified/corrected.
4. In Figure 1d the phase difference between the plateau regions is only ~100 deg, not 180. This implies incorrect calibration and should be re-measured.
5. P9 and Fig 2a,b: The description of what the red, blue and yellow dotted lines/regions denote is very hard to follow and understand. Particularly the phrase "are marked by blue and yellow dotted lines to denote bulk and surface regions, respectively" I cannot see how the blue and yellow dotted lines correspond to the bulk and surface regions, which are labelled on either side to the red dotted line. This should be made much more clear.
6. In parts of the narrative the authors have not included enough information to fully understand what has been done or to explain their results. These should be explained more clearly, including:
 - a. P10: "This was achieved by selectively growth of STO nanolayers on the PTO surface" At least some detail should be included on how this selective growth was achieved, as this is very important to understand how the structural improvement of the PTO surface/interface may have been achieved. This is too important to just leave to an experimental section (which isn't referred to anyway).
 - b. P11: "Consequently, the STO nanolayers selectively grown on the positive polarization facet led to negative polarization facet exposed" This is not easy to follow. The explanation should be made clearer.
 - c. P11: "Additionally, the results of photodeposition of Au and MnOx (Fig. S5) demonstrate consistent charge separation directions" The authors seem to be expecting us to read the SI to understand what they mean by this, but even the SI explanation is not clear. This should be explained briefly in the main paper.
 - d. P11: The authors briefly mention a Rh cocatalyst, but this is the first time it is mentioned and it is not discussed again until much later. So they should explain the context of this more clearly. Also, again, they overly rely on the SI to explain what they are talking about, when the SI is not very clear either.
7. P12: "Since presence of the defective structure near the surface of PTO is confirmed, the nature of the defects should be further elucidated" The authors have just been talking about the PTO/STO system, where they say there are not any defects on the PTO surface. Here I assume they are talking about bare PTO (which is the case for Fig 4c), but this is not very clear.
8. Figure 6d: The text states that there is something showing the relaxation of hot carriers in the "bottom left" of 6d. However, I can't see this. In general 6d is very hard to follow/understand and should be improved.
9. The decay curves in Fig S19 should be normalised so the kinetics can be more easily compared (probably show normalised plots as well as as-measured intensities).
10. The SI figures should be re-ordered so they follow the order of the main text more logically.

Reviewer #4

(Remarks to the Author)

Ferroelectric effects can potentially enhance solar energy conversion with photocatalysts. This manuscript evaluates this possibility for PbTiO₃ coated with SrTiO₃. The manuscript is interesting for the readership of Nature Comm. The quality of the Figures and of the text is very good. I recommend publication after a major revision:

1. The ferroelectric enhancement of overall water splitting is the main point of this work, but it is not directly demonstrated, only indirectly implied. The authors should include data to show that reversal of the polarization direction or heating above the Curie temperature will remove the ferroelectric enhancement of water splitting. See for example Assavachin, S. J. Am. Chem. Soc. 2023, 145(34), 18825-18833 and Morris, M.R., Advanced Materials, 2016, 28(33), 7123-7128.
2. The suppression of defects by the SrTiO₃ layer is a central theme in this work, but the defects are not chemically identified in the text. Please include additional data on the nature of the defects and explain why they cause charge recombination.
3. It is not clear if the charge carriers for water splitting are generated by excitation of the PbTiO₃ or by the SrTiO₃. The latter is a well known photocatalyst for overall water splitting. 100% quantum yield at 365 nm has been demonstrated. Takata, T., Nature, 2020, 581(7809), 411-414. Please include data to show PbTiO₃ is driving the water splitting process, not SrTiO₃.
4. At 360 nm illumination, why can charge carriers from SrTiO₃ be neglected?
5. A band energy scheme with band edges should be shown to discuss the possibility of electron AND hole transfer from PbTiO₃ to SrTiO₃.
6. Fig 3f shows the SrTiO₃ only covering part of the PbTiO₃ crystal. Is this correct? Please explain.

7. Masses and volumes for all reagents and solvents should be given to allow others to reproduce this work.
8. The term 'highly efficient' should be replaced with quantitative information
9. A reference needs to be added after '...established procedure.' On line 496
10. The authors should say how H₂ and O₂ were measured. Was a closed loop reactor used? If not, how was air contamination accounted for?
11. How were catalysts purified after photodeposition of Rh, CrOx, and CoOOH?

Version 1:

Reviewer comments:

Reviewer #1

(Remarks to the Author)

The authors provide here a detailed rebuttal letter with additional new measurements. The revision changed the interpretation of several aspects (e.g., transient optical measurements) and to some extent clarified some aspects. Although the manuscript has been improved as a result of this revision (and I appreciate the hardwork of the authors), I still keep my original opinion of the work that the presented hypotheses are not yet backed up with solid evidence yet. The provided evidence for improved catalytic activity in PTO/STO nanoplates is subtle and not fully coherent in my view in the revised manuscript.

Reviewer #2

(Remarks to the Author)

At this stage, the authors improved the analysis, which support their claims. I have no objection for the article publication at this step.

Reviewer #3

(Remarks to the Author)

I thank the authors very much for their response to my original comments. The majority of the comments have been fully addressed. In particular, the most important point (#1) has been addressed both in the response, and through the clarification in the text. There are just a couple of outstanding minor points that have not been fully addressed, or addressed incorrectly. Once these are corrected, I am happy for the paper to be accepted. They are:

1. Q3 in rebuttal: this has not been correctly changed. The references refer to ferroelectric memories. The phrase should be changed to 'ferroelectric memories' not 'ferroelectric memorizers'.
2. The authors have misunderstood my point, and appear to have manually shifted the y-axis of the phase signal. This should not be done, as a y-offset can be an important feature in a PFM phase loop. However, this was in part because I made a mistake in my original comment, for which I apologise. I misread the axis, and the original phase difference was indeed close to 180 degrees. Therefore the authors should re-instate the original version of the figure.
3. Q7 in rebuttal: where the authors have added the text "The growth of STO on PTO was induced by the electrostatic adsorption of STO precursors onto the charged STO surface." Should this read "...onto the charged PTO surface"?

Reviewer #4

(Remarks to the Author)

The authors have responded satisfactorily to most of my concerns. There are some residual concerns, that can be addressed in a minor revision.

1. OK
2. OK
3. Authors mention the results of control experiments with STO PTO mixtures, but the data is not shown.
4. We know from work by Rohrer and coworkers, that ferroelectric enhancements of SrTiO₃ can result from direct contact between a ferroelectric and SrTiO₃. See for example <http://dx.doi.org/10.1021/cm1018025> Mixing SrTiO₃ with PbTiO₃ particles does not correctly test the hypothesis that charge carriers can be generated by excitation of the SrTiO₃ coating
5. No band energy scheme was provided. Adding such a scheme would greatly benefit the understanding of this work.
6. OK
7. OK
8. OK
9. OK
10. Based on the revised text, it is still not clear if a closed-loop reactor was used or not. For this type of experiment, a closed loop is needed to avoid false readings from air contamination.
11. OK
12. New query: Why is anodic photocorrosion (Pb²⁺ to Pb⁴⁺) not a problem in this system?

Version 2:

Reviewer comments:

Reviewer #1

(Remarks to the Author)

I don't have further comments.

Reviewer #4

(Remarks to the Author)

This looks good now. I recommend publication as is.

Unveiling Charge Utilization Mechanism in Ferroelectric for Water Splitting

I am submitting the revised manuscript entitled “**Unveiling Charge Utilization Mechanism in Ferroelectric for Water Splitting**” (No. NCOMMS-24-31509-T) for your consideration of publication in Nature Communications. We are very pleased to hear that the reviewers felt that we have presented interesting and innovative findings. We want to extend our appreciation for taking the time and effort necessary to provide such guidance. We have carefully considered the comments from the reviewers and addressed these comments in the revised copy. The following are the point-by-point responses to the comments with a description of the changes made in the revised version of the manuscript.

Contents

Referee 1	Pages 3–13
Referee 2	Pages 14–36
Referee 3	Pages 37–52
Referee 4	Pages 53–62

Reviewer #1 (Remarks to the Author):

The authors use here a large suit of characterization techniques to probe the mechanisms why photocatalytic activity of the PTO improves when coated with thin layers of STO. The first part of the manuscript is well written. The results and discussion part is somewhat cluttered due to use of many techniques and the results being shown in short snippets to justify the hypothesis of the authors. However, I have multiple reservations on the data presented hence find the validity of the presented results questionable in the presented format.

Q1. Figure 1d (bulk region) and Figure 4b P1 (in the bulk region) does not seem to agree and e_g feature is missing in P1 region.

Response:

Thank you for your valuable suggestions regarding the interpretation of the EELS data. In order to improve our understanding of the data, we have conducted higher resolution EELS measurements. These measurements reveal a distinct e_g and t_{2g} feature with noticeable d-orbital splitting of Ti within the oxygen octahedral field in the PTO region. Additionally, both the bulk phase of pure PTO and the PTO-STO samples exhibit consistent splitting patterns in the EELS data, further supporting our analysis.

To enhance the clarity and interpretability of the EELS data, we have revised the corresponding figure to facilitate a more straightforward comparison of the changes observed after the growth of STO nanolayers. Specifically, we have relocated the EELS data from Fig. 2 and Fig. 4 to Fig. S4 and Fig. S15, respectively, where they are now presented with improved resolution. Furthermore, we have incorporated a direct

comparison of the surface EELS spectra of PTO before and after STO growth in Fig. 4c.

Furthermore, we have corrected a mistake in the previous manuscript related to the identification of the t_{2g} and e_g peaks in the EELS spectrum. Previously, we noted the e_g at a lower energy position and t_{2g} at a higher position by mistake. In the Fig. 4c, we have corrected it. Additionally, for easier comprehension of the splitting of d orbitals, a schematic was added in Fig. 4d to explain the splitting of d orbitals in different structure of oxygen octahedron field.

Consequently, the corresponding sentence located in line 279, page 14 has been revised to reflect this accurate identification. 'These changes indicated a reduction in defect density and a transition to a more ordered interface between PTO and STO. The peaks splitting could be explained by the schematic in Fig. 4d. In the cubic phase of STO, Ti atoms are located at the center of oxygen octahedron. In this configuration, the Ti d-orbitals split into t_{2g} (d_{xy} , d_{xz} , d_{yz}) and e_g ($d_{x^2-y^2}$, d_z^2) levels. In contrast, in the tetragonal phase of PTO, the elongation along z axis and contraction in the x and y axes reduced the symmetry. This results in Ti being displaced from the center of the oxygen octahedron, further reducing degeneracy along the x, y, and z directions. Consequently, the t_{2g} and e_g orbitals underwent further splitting in each peak and exhibited less splitting between t_{2g} and e_g peaks. At defective regions of PTO and STO surfaces, further distortion may exacerbate the reduction in degeneracy along the x and y directions, resulting in increased splitting in t_{2g} orbitals and a reduction in the splitting between t_{2g} and e_g peaks. This explained the observed differences in peak splitting

between defective PTO and STO surfaces.'

Fig. 4 c, EELS data of PTO surface before and after growth of STO nanolayers. **d**, schematic of splitting of d orbitals of Ti in oxygen octahedron field. (The red, blue, yellow and green balls represent Sr, Pb, Ti and O, respectively.) **d**, schematic of splitting of d orbitals of Ti in oxygen octahedron field. (The red, blue, yellow and green balls represent Sr, Pb, Ti and O, respectively.)

Fig. S4 Surface structure evolution in positive polarization of PTO. **a**, STEM data of PTO surface. **b**, EELS evolution from bulk to surface of PTO.

Fig. S15 Surface structure evolution in the interface of PTO. **a**, STEM data of PTO-STO interface. **b**, EELS evolution from PTO to STO.

Q2. Negative control group reference measurement of 10:1 PTO and STO mixture shows a much stronger 20 $\mu\text{mol/h}$ H_2 production rate vs. only PTO showing 3.27 $\mu\text{mol/h}$. This represents a 6-fold improvement without any treatment. Thus, bare PTO may not a realistic comparison point, hence 400-fold improvement may be an over estimation.

Response:

We have amended the description in the abstract section of the **revised** manuscript to eliminate any potential for misunderstanding, stating: ‘As a result, the apparent quantum yield at 365 nm (AQY) for the overall water splitting activity was significantly increased from 0.01% for bare PTO and 0.2% for PTO mixed with STO to an unprecedented 4.08%, representing a nearly 400-fold and 20-fold enhancement, respectively’ in line 44, page 2.

Q3. Figure 4 is misleading. X-axis scale in Figure 4c is very narrow whereas in Figure

4d it is much wider range. Also, y-axis is not specified in values. Thus, Figure 4c vs. Figure 4d comparison is currently not meaningful.

Response:

We have updated the TOF-SIMS data and changed the Y-axis unit to the universal MS unit labeled 'counts'. Further, we calibrated the depth, as shown in Fig. S13, and changed the X-axis to a more visually intuitive length unit.

The revised figures are now in Figure 4a and 4b.

Fig. S13 Calibration of the depth of PTO and STO during TFO-SIMS tests. **a,b**, element evolution of PTO single film and STO single substrate with sputter time, respectively. **c,d**, depth of sputter region after sputter.

Fig. 4 a,b, shows the element distribution profile across the positive polarization direction before and after the growth of STO by TOF-SIMS, respectively.

Q4. Some of the authors from this work have studied the same PTO/STO system for its photocatalytic activity Ref. 44. There, they reported maximum H₂ production rate up to 80 $\mu\text{mol/h}$ whereas the obtained value is 200 $\mu\text{mol/h}$ in the current work as a result of STO thickness optimization. It is not clear the hypotheses proposed by Ref. 44 can be refuted by the current work on the mechanism of photocatalytic activity improvements.

Response:

The hypotheses proposed by Ref. 44 didn't have conflict with the opinion we proposed because of the different interface structure under different prepare condition of both PTO and the grown STO. From the EPR data listed below, the interface defects were not effectively eliminated in Ref. 44. However, in our case, the interface defects were almost eliminated. These differences in surface structure could potentially influence water splitting.

In particular, PTO prepared in Ref. 44 was synthesized under 6 M KOH at 200 °C, whereas we prepared PTO under 2 M KOH at 260 °C. Similarly, STO growth in Ref.

44 occurred under 4 M NaOH, whereas we grew STO using 2 M NaOH. These variations in preparation conditions likely contributed to the differing interface structures observed.

EPR spectra of single-domain PbTiO₃ nanoplates, PbTiO₃/L-SrTiO₃, PbTiO₃/M-SrTiO₃ in Fig. 2a of **Ref .44**.

EPR data in our manuscript could be seen in page 19 in Supplementary Information materials, Fig. S17.

In order to express clearly the feature of our sample, we revised the corresponding paragraphs in the revised manuscript: In line 209, page 19 in Supplementary Information, ‘After growing STO nanolayers, electron paramagnetic resonance spectroscopy (EPR) peaks dramatically decreased, indicating a decrease in the concentration of defects. In particular, the defect peaks in the PTO-STO samples were only slightly larger than the fluctuation of the EPR baseline, further confirming the presence of fewer defects.

It is worth noting that the PTO samples prepared in this work were subjected to higher temperatures than those in similar studies.¹ Typically, higher temperatures result in increased formation of defects in our PTO samples. However, contrary to this

expectation, we found that the defects significantly reduced after the growth of STO nanolayers. We hypothesize that this reduction in defects may be related to the different concentration of NaOH used during the preparation process. Specifically, the NaOH concentration we used here was 2 mol/L, which is half of that used in a recent report. This difference in concentration could have contributed to the observed reduction in defects.'

Fig. S17 Defects elimination of PTO after growing STO nanolayers.

Q5. Overall, the scientific approach of the authors is rather qualitative. The logical inferences are made based on changes, and sometimes subtle changes, in the experimental data (Ex./ EELS measurements). Thus, the claims of the authors are not clear nor convincing at times.

Response:

We apologize for the previous low-quality data and the resulting unclear conclusions. With the valuable advise from you and other reviewers, we have conducted additional experiments with higher-quality and clearer results, particularly through EELS and

TOF-SIMS data.

As highlighted in our response to **Q1**, the EELS data provide clear evidence of structural evolution in both bare PTO and PTO-STO samples. In bare PTO, the decreased splitting between the t_{2g} and e_g peaks in the surface region indicated the presence of surface defects. In contrast, in PTO-STO samples, the increased splitting between the t_{2g} and e_g peaks at the interface region confirmed the elimination of Ti defects and demonstrated surface structure changes after STO modification. This confirms the structural transformation brought about by STO growth.

Additionally, we performed depth calibration in the TOF-SIMS data, as mentioned in our response to **Q3**. This allowed us to directly observe the distribution of Ti defects with depth in bare PTO, and after STO growth, the elimination of Ti defects became evident. These improvements strengthen the credibility of our claims regarding surface structure modification and defect elimination with STO growth.

Q6. Transient mid-IR probe experiments are somewhat misleading. The assignment of the authors that the signal flip indicates long-lived bleach needs further support by wavelength dependent measurements. The effect may very well arise from some sort of interfacial small energy state. I do not buy the argument on impact ionization. That seems unlikely unless the authors show this carefully with excitation fluence and pump energy dependent studies.

Response:

Thanks for your valuable suggestions. We concur with your deduction regarding the impact of interfacial small energy states. Upon re-conducting the experiment with

different probe wavelengths, we observed that an increase in wavelength led to the transformation of the bleaching signal into an absorption signal, providing evidence for the existence of these small energy states. Additionally, the persistence of the long-lived bleaching signal further reinforced our hypothesis that electron lifetime can be extended through STO nanolayer modification.

We conducted a series of experiments using varying probe wavelengths and discovered that when we shifted to a larger wavelength, the bleaching signal transformed into an absorption signal, as illustrated in Fig. S25. This observation supports your deduction that small energy states were involved in the signal flip. Specifically, the bleaching signal originated from these small donor states, exhibiting similarities to the ground state bleaching signal. However, the larger wavelength was inadequate for exciting electrons within these states.

In light of these new experimental findings, our previous attribution of the increased bleaching signal to impact ionization now appears to be unreasonable. Nevertheless, the enduring bleaching signal continues to uphold our hypothesis that electron lifetime can be prolonged following modification with STO nanolayers.

Since this portion of the data does not provide additional insights beyond other characterization methods, such as TPV, we have relocated the MIR-TAS data to Fig. S25. Consequently, we have amended the description in the main text and the schematic related to "impact ionization" accordingly.

In lines 301, page 28, the original sentences of supplementary Information 'In Fig. S25a, both bare PTO and STO exhibited absorption signals in MIR-TAS results, while

PTO-STO showed a long-lived bleaching signal. To investigate the formation of the bleaching signal in the PTO-STO sample, the probe light wavelength was varied. As shown in Fig. S25b, the PTO-STO sample exhibited a bleaching signal at probe wavelengths of 3400 nm and 4000 nm. However, when the probe light wavelength was increased to longer than 6000 nm, the signals transitioned to absorption, resembling the shift from ground state bleaching (GSB) to normal absorption observed when the probe wavelength changes from shorter to longer. This indicated that interface states were formed during STO growth. Short-wavelength light could excite these interface states, resulting in a bleaching signal, whereas longer-wavelength light was insufficient to excite these states and thus showed absorption' were revised to description of the interface states after growth STO on PTO.

Fig. S25 a, MIR-TAS of PTO, STO, PTO-STO samples. **b**, MIR-TAS of PTO-STO sample with different probe light. **c**, Normalized MIR-TAS of PTO-STO sample with 3400 and 4000 nm probe light.

Reviewer #2 (Remarks to the Author):

The article deals with interesting questions regarding the reason behind the low activity of ferroelectric PbTiO_3 . It is suggested that low activity is due to the presence of Ti defect at the surface. The authors proposed healing these defects by covering PTO with STO layer. The authors attempt to investigate the interfacial Ti^{3+} defect using EELS. Despite the fact that the analysis shows some differences, the conclusion cannot be strongly supported. In fact, the main results that should confirm the assumption of the 3d splitting into t_{2g} and e_g is not fully convincing. The EELS spectra show that the splitting became lower from bulk to the STO interface, as it was observed for EELS results from the PTO bulk to the PTO surface. In addition, there are many results that are controversial, like the results obtained by TOF-SIMS and EDS. The experimental section is not sufficiently described for the reader, and many information is missing. The results obtained in this article cannot support either the conclusion or the main question raised at the beginning. For all these reasons, I cannot recommend this article for publication.

Bellow more recommendations and comments:

Q1. Despite the fact that the analysis shows some differences, the conclusion cannot be strongly supported. In fact, the main results that should confirm the assumption of the 3d splitting into t_{2g} and e_g is not fully convincing. The EELS spectra show that the splitting became lower from bulk to the STO interface, as it was observed for EELS results from the PTO bulk to the PTO surface. There are many results that are controversial, like the results obtained by TOF-SIMS and EDS. The experimental

section is not sufficiently described for the reader, and many information is missing.

Response:

Based on your questions and the suggestions from other reviewers, we have conducted additional experiments to further enhance the interpretation of the data. In particular, we focused on analyzing the splitting of e_g and t_{2g} peaks in the Ti L_2 and L_3 peaks.

Our new findings show that there is a clear evolution of these peaks, and importantly, there are no conflicting results between the new data and the previous data. Specifically, we observed a decreased splitting between the t_{2g} and e_g peaks in the surface region of PTO, which is consistent with the presence of defects in this region. This observation was further supported by TOF-SIMS analysis, which showed similar results.

Additionally, we analyzed PTO-STO samples and found an increased splitting between the t_{2g} and e_g peaks from the PTO bulk to STO. This finding is consistent with the elimination of surface defects in PTO, indicating that the changes in peak splitting are related to the presence or absence of defects.

Overall, these new results provide valuable insights into the interpretation of the data and further strengthen our understanding of the phenomena observed in the PTO and PTO-STO samples.

Q2. Lack of fair and detailed procedure in the article. At this step, the experimental part cannot be reproduced correctly. Important details in the experimental part are missing:

Page 9: “Fig. 2a and Fig 2b illustrate the cross-section of PTO” How was the sample prepared to check the cross section in STEM?

Response:

The PTO samples shown in Fig. 2a were prepared using a physical slicing method with a diamond blade, in order to circumvent potential defects that may arise from alternative methods such as FIB.

This prepare method have been added in the section of Structure characterization in line 668, Page 38. A revised content has been added to describe the prepare method:

‘STEM and EELS sample was prepared through by physically slicing with a diamond blade. Briefly, the particles were embedded in the resin, after resin had hardened, the head was cut to approximately 150×200 nm using a diamond blade. Subsequently, the head was sliced to approximately 50 nm sections with the diamond blade. Finally, the slices were transferred onto a copper mesh for testing.’

Q3. In the same line, details of the synthesis process and the photocatalytic test are missing. The experimental part should be complemented and detailed:

for example, what was the volume of the water? What is the material of the vessel used? Quartz? sodocalcique glass? Pyrex? What was the volume of the reactor? How was H_2 calculated depending on the dead volume of the reactor? Was it pure water? What was the flux reaching the sample during the photoreaction?

Response:

We have added detailed information on the synthesis process and photocatalytic testing. Briefly, the volume of the water used during the test was 100 mL. The reaction

was conducted in a high borosilicate glass vessel with a quartz cover. The volume of the reactor and its dead volume were not critical, as we used external standard quantification to measure the amounts of H₂ and O₂ produced. The water we used was DI water with a resistance of 18.2 MΩ.

The photon flux reaching the samples is shown in Fig. S30.

Fig. S30: a, Photon flux of 300 W Xe lamp under photocatalysis. b, Photon flux of 300 W Xe lamp with 365 nm band-pass filter.

Some detailed revision in experiment supplement

In line 533, page 32 the method of prepare PTO could be seen as follow: PbTiO₃ single domain crystals were prepared by hydrothermal method, following a previously established procedure. 0.2396 g Anatase TiO₂(Alfa), 1.2421 g Pb(NO₃)₂(Aladdin) and 3.36 g KOH(Aladdin) were added in 5 mL, 10 mL and 15 mL deionized water respectively. The resulting solutions were then transferred to a 50 mL Teflon-lined reactor, briefly stirred, and heated at 260 °C for 36 hours. After reaction, the yellow powders were washed 3 - 5 times with deionized water and dried at 80 °C.

In line 541, page 33 the method of prepare PTO-STO could be seen as follow: 2.4

g NaOH(Aladdin) was dissolved in 15 mL DI water, and then 3 mL 0.1 M tetrabutyltitanate(dissolved in 1 M HNO₃) was added in NaOH with stirring. After that, 12 mL DI containing 0.0635g Sr(NO₃)₂ was added to the mixture. Finally, 0.909 g prepared PbTiO₃ was added. After a 30 min stirring, the mixture was transferred to a 50 mL Teflon-lined reactor and heated at 200 °C for varying reaction times. After reaction, the resulting products were washed 3–5 times with a combination of ethanol and water and subsequently dried at 80 °C.

In line 560, page 33 the method of Photocatalysis process: The photocatalytic reaction was conducted by placing the high borosilicate glass reaction vessel, with a quartz cover and containing 100 mL of deionized water, in a water bath at 15 °C. The reaction was irradiated with a 300 W Xe lamp. A total of 200 mg of the photocatalyst, with cocatalyst loaded in situ without further purification, was used. Sampling for product analysis via GC was performed at one-hour intervals. After each sampling, the reaction vessel was subjected to vacuum for 10–15 minutes. The entire reaction process was carried out using the Perfectlight Sci&Tech reactor. For quantifying H₂ and O₂ production, external standard quantification was employed. Different amounts of H₂ and O₂, in a 2:1 ratio, were successively injected into the reactor containing 100 mL of water. After allowing 30–60 minutes for equilibrium, the gas was injected into the GC with an Ar carrier. A series of data were acquired, and the plot of the injected amounts of H₂(O₂) versus the peak areas provided the reactor's calibration coefficient. During the test, the amounts of H₂ and O₂ production were calculated using this coefficient and the peak areas of H₂ and O₂.

Q4. Piezoresponse force microscopy (PFM). What was the adherent used for the PTO to stick in Au film?

Response:

PTO particles was embedded in the Au film. The prepare procedure could be found in Fig S31.

Fig. S30: a, Schematic of PFM Sample Preparation Procedure

In line 613, page 35, ‘The preparation of PFM samples was carried out using the following method: First, the particles were dispersed on a monocrystal polished Si surface. Next, a layer of Au film was evaporated onto the Si surface. Subsequently, the Au film was adhered to a conductive ITO surface using an adhesive layer. Finally, the Si substrate was peeled away, transferring the particles and Au film onto the ITO substrate. Before testing, electrical contact was established by connecting the Au film and exposed ITO with conductive Ag adhesive’ were **revised** to show the PFM sample preparing.

Q5. NO details are reported in the ms regarding the acquisition of EELS analysis: what are the pixels of the line? Pixel size? Channel? Etc... This step is crucial since it is known that depending on how the samples are prepared, in EELS, the Ti^{3+} signal comes

from the damaged foil surfaces.

Response:

During EELS test the channel was set to 0.15 eV/channel and the pixel size was $0.5 \text{ nm} \times 0.5 \text{ nm} \times 0.15 \text{ eV}$, the pixels during PTO samples, the pixel dimensions were 195×215 , and for the PTO-STO samples, they were 62×162 . The STEM and EELS samples were prepared using a physical slicing method with a diamond blade. Specifically, the particles were embedded in resin, and after the resin hardened, the block was cut into approximately $150 \times 200 \text{ nm}$ pieces with a diamond blade. These pieces were then sliced into approximately 50 nm sections, which were transferred to a copper mesh for testing. This method minimizes damage from the ion beam.

This **revision** has been added to the Materials Characterization section of the experimental in line 665, page 38.

Q6. The degeneracy in the case of Ti^{4+} in bulk PTO, which is reported to be partially lifted by the octahedral crystal field, separating 3d into t_{2g} and e_g , e.g., levels, is weak compared to reported studies. No reasonable conclusion can be made from these results since one can argue that it is a simple effect of the Ar ion polishing usually used for the sample preparation for such kind of measurement. The EELS analysis is a bit superficial and should be complemented by an extraction of the Ti^{3+} ratio and compared for different samples using variable preparation.

Response: Previous EELS data were obtained at low resolution; higher resolution data are now provided in Fig. S4 and Fig. S15. PTO preparation for STEM was using physical slicing method, without polishing by ions, so the defects couldn't from the

sample preparing. Additionally, we clarified that the Ti defects mentioned refer to Ti point defects within the unit cell rather than Ti^{3+} , which typically forms due to oxygen vacancies.

Fig. S4 Surface structure evolution in positive polarization of PTO. **a**, STEM data of PTO surface. **b**, EELS evolution from bulk to surface of PTO.

Fig. S15 Surface structure evolution in the interface of PTO. **a**, STEM data of PTO-STO interface. **b**, EELS evolution from PTO to STO.

Q7. In addition, it is speculated that STEM analysis shows defect localization. It is

surprising how the author comes to such a conclusion.

Response:

ADF-STEM data illuminated the atom position in the crystal as well as the unit structure. We could see the apparent distortion in the surface region in Fig. 2, as results of breaking translational symmetry.

A Fast Fourier Transform (FFT) was conducted on the images from both the bulk and surface regions of PTO for a clearer demonstration.”

Fig. 2 Surface structure of PTO. **a,d**, STEM of PTO near positive polarization surface, the blue and yellow dot lines are the atom row of PTO in bulk and surface, respectively. The red dot line refers to the bulk and the surface bound. **b**, schematic of unit structure in PTO cell, the blue, yellow, and green balls represent Pb, Ti and O atoms, respectively. The blue arrow points the direction of polar in PTO unit cell, which is same as the shift direction of Ti atom in unit cell. **c**, magnified HADDF STEM of PTO in magenta region of **a**. blue and yellow balls represents Pb and Ti atoms, respectively. **e,f**, showed the FFT results of bulk and surface region, respectively. The circle represents the (101) direction of PTO, and the angle was the angle between the (101) point and the x axis.

In line 178, page 9, 'Fig. 2a and Fig. 2d illustrated the cross-section of PTO near the positive polarization surface. The polarization direction could be determined by the Ti atom shift within the unit cell. As depicted in Fig. 2b, the schematic illustrated the upward polarization of unit cell in ferroelectric tetragonal PTO, where the Ti atom shifts upward along the z-axis, while the oxygen octahedron shifts downward, generating an upward polarization within the unit cell. In the magnified image in Fig. 2c, Ti atom is shown to shift upward, indicating an upright polarization direction. Consequently, the surface in Fig. 2a can be confirmed as having a positive polarization direction. In Fig. 2a the red-dotted rectangle highlights the near-surface region on the positive surface of PTO. In Fig. 2d, the surface and bulk regions were delineated by the red dotted line. The angle of (101) direction in the bulk region was 41.56° relative to the x axis, while at the surface, it changed to 43.09° , indicating surface region distortion' were revised to verify the existence of the surface defects.

Q8. Once the authors analyzed the separating 3d into t_{2g} and e_g , the splitting became lower, as was observed by their analysis in Fig. 2 from the bulk to the surface. So why, in one case, is it due to surface defects, and in the other case, it is attributed to defect healing? It is a bit contradictory how it is justified.

Response:

Different tendency of the splitting extent between t_{2g} and e_g showed the different change of surface structure. In bare PTO, the splitting in surface region become less while it became larger in after modification with STO. So, there weren't contradiction between these results.

The surface defects were identified based on both the distortion of unit cells in the surface region and the decreased splitting of the L_3 peaks of Ti in this region. Point defects in the crystal decrease the translational symmetry and induce distortion. From Fig. S4 and Fig. S15, the EELS data reveal that in bare PTO, the splitting of the L_3 peaks decreases from the bulk to the surface region. This decreased splitting between t_{2g} and e_g can be ascribed to the decreased degeneracy of the d orbitals in unit cell of PTO causing by the defects. While in PTO-STO the splitting between t_{2g} and e_g showed opposite results that increased in the surface region, even close to in STO. This was attributed to defect healing and formation of a transition structure.

For a clearer comparison, we moved the previously EELS from Fig. 2 and Fig.4 to Fig. S4 and Fig. S15. Additionally, we add the direct comparison EELS data of PTO surface before and after growth STO in Fig. 4c and the schematic to discuss the splitting of d orbitals in different unit cells.

The explanation for the splitting of the Ti 3d orbitals in the oxygen octahedral field has been added to Fig. 4d. The explanation is as follows:

In line 277, page 14, 'Fig. 4c showed that the surface structure of PTO, initially characterized by defective regions with minimal splitting of the t_{2g} and e_g peaks, transforms to exhibit more pronounced peak splitting after STO modification. These changes indicated a reduction in defect density and a transition to a more ordered interface between PTO and STO. The peaks splitting could be explained by the schematic in Fig. 4d. In the cubic phase of STO, Ti atom is located at the center of oxygen octahedron. In this configuration, the Ti d-orbitals split into t_{2g} (d_{xy} , d_{xz} , d_{yz}) and e_g ($d_{x^2-y^2}$, d_z^2) levels. In contrast, in the tetragonal phase of PTO, the elongation along the z-axis and contraction in the x- and y-axes lead to a reduction in symmetry. This, in turn, causes Ti to be displaced from the center of the oxygen octahedron, further lifting the degeneracy along the x, y, and z directions. Consequently, the t_{2g} and e_g orbitals performed further splitting in each peak and exhibited less splitting between t_{2g} and e_g peaks. At defective regions of PTO and STO surfaces, further distortion may exacerbate the reduction in degeneracy along the x and y directions, resulting in increased splitting in t_{2g} orbitals and a reduction in the splitting between t_{2g} and e_g peaks. This explained the observed differences in peak splitting between defective PTO and STO surfaces.'

were **revised** to explain the EELS data.

Fig. 4 c, EELS data of PTO surface before and after growth of STO nanolayers. **d**, schematic of splitting of d orbitals of Ti in oxygen octahedron field. (The red, blue, yellow and green balls represent Sr, Pb, Ti and O, respectively.)

Q9. In Fig. 2, it is speculated that the positively polarized surface is investigated. However, there is no mention of how positively or negatively charged surfaces are discriminated.

Response:

We determined the polarization direction based on the shift of Ti atoms within the unit cell. Specifically, the positive polarization facet was confirmed using ADF-STEM data. To enhance clarity regarding the charge separation direction, we incorporated a unit cell structure into **Fig. 2b**, which clearly illustrates the polarization direction in PTO. Furthermore, we included a magnified STEM image in **Fig. 2c** to better visualize the displacement of Ti atoms within the unit cell.

By identifying the positions of both Pb and Ti atoms, we were able to infer the polarization direction from the shift of Ti atoms (as depicted in **Fig. 2b**). In **Fig. 2c**, where Pb atoms are represented by large blue balls and Ti atoms by small orange balls,

we positively identified the surface orientation. It is worth noting that this method for determining polarization direction has been previously utilized in the literature. (eg. C. L. Jia, K. W. Urban, M. Alexe, et al., *Science*, 2011, 331, 1420).

In line 277, page 14, 'Fig. 4c showed that the surface structure of PTO, initially characterized by defective regions with minimal splitting of the t_{2g} and e_g peaks, transforms to exhibit more pronounced peak splitting after STO modification. These changes indicated a reduction in defect density and a transition to a more ordered interface between PTO and STO. The peaks splitting could be explained by the schematic in Fig. 4d. In the cubic phase of STO, Ti atoms are located at the center of oxygen octahedron. In this configuration, the Ti d-orbitals split into t_{2g} (d_{xy} , d_{xz} , d_{yz}) and e_g ($d_x^2-y^2$, d_z^2) levels. In contrast, in the tetragonal phase of PTO, the elongation along z axis and contraction in the x and y axes reduced the symmetry. This results in Ti being displaced from the center of the oxygen octahedron, further reducing degeneracy along the x, y, and z directions. Consequently, the t_{2g} and e_g orbitals underwent further splitting in each peak and exhibited less splitting between t_{2g} and e_g peaks. At defective regions of PTO surfaces, further distortion may exacerbate the reduction in degeneracy along the x and y directions, resulting in increased splitting in t_{2g} orbitals and a reduction in the splitting between t_{2g} and e_g peaks. This explained the observed differences in peak splitting between defective PTO and STO surfaces.' were revised to explain the evolution of the EELS data at different structure.

Fig. 2 Surface structure of PTO. **a**, STEM of PTO near positive polarization surface, the blue and yellow dot lines are the atom row of PTO in bulk and surface, respectively. The red dot line refers to the bulk and the surface bound. **b**, schematic of unit structure in PTO cell, the blue, yellow, and green balls represent Pb, Ti and O atoms, respectively. The blue arrow points the direction of polar in PTO unit cell, which is same as the shift direction of Ti atom in unit cell. **c**, magnified HADDF STEM of PTO in magenta region of **a**. blue and yellow balls represents Pb and Ti atoms, respectively.

Q10. The analysis of the oxidation state of Ti should follow a deep analysis of the ratio of Ti^{3+} from bulk to the interface in PTO and PTO/STO.

Response:

We apologize for any confusion regarding the Ti-related defects. In this study, the Ti defects we focused on were point vacancies within the unit cell, rather than Ti^{3+} ions. It's important to note that the formation of Ti^{3+} is usually linked to oxygen vacancies. Therefore, by examining the distribution of Ti atoms from the surface to the bulk, we were able to infer changes in the Ti point vacancies.

Q11. Fig. 4, it is speculated that the distortion disappeared. However, more analysis and images of different regions of variable PTO-STO particles should be provided to come to such a conclusion. In fact, the evolution of the EELS spectra from P1 to P5 is very similar to the one recorder in PTO from bulk to the surface, making the conclusion questionable.

Response:

Repetitive experiments with other particles and higher resolution analysis were conducted, as shown in Fig. S31 in the revised manuscript. Similar results regarding the interface structure were observed, which verifies the reliability of our findings: after modification with STO nanolayers, surface defects in PTO were effectively eliminated.

In line 417, page 42 of Supplementary Information, ‘The PTO-STO interface was examined using additional particles. As shown in Fig. S32, a well-structured interface was observed, consistent with the results in Fig. 3c, further verifying the reliability of our findings that, after modification with STO nanolayers, surface defects in PTO were effectively eliminated’ were **revised** to verify our findings.

Fig. S32: repeat results of PTO-STO interface.

Q12. The TOF-SIMS profile shows that Ti concentration increases gradually when recording the signal from the surface of STO, and then it becomes constant in PTO. The decrease should be expected in PTO extreme surface and not in the STO. This is surprising and in contradiction with the EELS conclusion, and the EDS analysis. Also, the unit of depth axis is wrong, it should be “nm” (or another unit) and not a.u. The intensity should be “counts”.

Response:

Firstly, we correct the unit of intensity to ‘counts’ and the unit of depth to ‘nm’.

Secondly, we would like to emphasize that there is no contradiction between the EELS data and the EDX data. The main reason is that the quantification in the EDX data was not entirely accurate due to variations in thickness across different regions of the sample. These samples were prepared by physically slicing them with a diamond blade using a rotating mechanical arm, which led to non-uniform slice thicknesses. This inconsistency is evident in the similar trends observed in the O intensity across the interface, as illustrated in Fig. S16.

To show the reason for the discrepancy between TOF-SIMS DATA and the EDX data, we revised the corresponding paragraphs: In line 198, page 17 of Supplementary Information, ‘It is important to note that the apparent increase in Ti intensity from the PTO interface to the bulk seemingly conflicts with the TOF-SIMS data. This discrepancy arises because the EDS measurements, conducted using STEM, likely captured increased signals due to non-uniform sample thickness caused by the slicing process. The samples were prepared with a diamond blade mounted on a rotating arm,

resulting in arc-shaped sections with varying thickness, thus affecting the quantitative data accuracy.

Fig. S16 EDS results near PTO-STO surface. The bar in the SEM was 10 nm.

Additionally, we have updated the units in our TOF-SIMS data to provide a clearer representation in **Fig. 4a and 4b**. We also calibrated the sputter time with the depth of PTO and STO, as shown in **Fig. S13**.

Fig. S13 Calibration of the depth of PTO and STO during TFO-SIMS tests. **a,b**, element evolution of PTO single film and STO single substrate with sputter time, respectively. **c,d**, depth of sputter region after sputter.

Fig. 4 **a,b** shows the element distribution profile across the positive polarization direction before and after the growth of STO by TOF-SIMS, respectively.

Q13. Figure S4 is neither discussed nor mentioned in the main manuscript.

Response:

Fig S4 was moved to Fig. S6, and flowing description was added:

In line 207, page 11, ‘Meanwhile, the slight increase in absorption below 350 nm indicated the nearly unchanged light absorption properties after STO modification (Fig. S6)’ were revised to explain the neglectable influence in light absorption after growth of STO.

Q14. How apparent quantum yield (AQY) was calculated?

Response:

The calculation of the AQY can be found in the supplementary materials. The number of incident photons is detailed in Fig. S28, the distribution of photons is listed in Table S2, and the activities under 365 nm illumination are presented in Table S3. The calculation for the number of incident photons is:

$$\text{Number of incident photons} = \sum_{r=0}^R f(r) \times \pi \times (r^2 - (r - \Delta r)^2)$$

The AQY calculation is:

$$\begin{aligned} \text{AQY} &= \frac{\text{Number of reacted electrons}}{\text{Number of incident photons}} \times 100\% \\ &= \frac{\text{Number of evolved H}_2 \text{ molecules} \times 2}{\text{Photon Flux} \times S \times t} \times 100\% \end{aligned}$$

Q15. The discussion justifying the decrease of the photoactivity for higher STO thickness is a little weak. The change in the space charge layer at the PTO/STO junction is only related to the potential of the flat bands in each oxide. The increase in the thickness should affect charge diffusion to the surface instead. The discussion should be reviewed in depth and correlated to the reduction of Ti in STO, which could

rationalize the decrease in the activity.

Response:

Thanks for your advice of the STO thickness dependent activities. As you mentioned, increasing the thickness of STO enhances the electron diffusion length within STO. The reduced Ti in STO bulk made less electrons transfer to surface and suppress the activities.

In line 344, page 17, ‘Additionally, it has been observed that some Ti defects also existed in STO layer. As the thickness of the STO layer on the PTO surface increases, electrons would diffuse over a greater distance within STO film. This extended diffusion length would increase the charge recombination probability through defects in STO layer, which in turn resulted in the decrease activities for water splitting’ were revised to explain the decreased activities with STO film.

Q16. The interpretation of the TRABS analysis is quite confusing. The authors attributed the bleaching signal to a shallow trap. The bleaching is usually attributed to the relaxation of electrons to the ground states. The authors are making conclusions about the presence of shallow traps with no proof. The original spectra of TRABS should show. If there are any traps, they should appear by new absorption bands.

Response:

Thanks for the advice, the similar questions have been raised by referee 1. We have made corrections to this part:

We attributed the bleaching signals to the impact of interfacial small energy states. Upon re-conducting the experiment with different probe wavelengths, we observed that

an increase in wavelength led to the transformation of the bleaching signal into an absorption signal, providing evidence for the existence of these small energy states. Additionally, the persistence of the long-lived bleaching signal further reinforced our hypothesis that electron lifetime can be extended through STO nanolayer modification.

We conducted a series of experiments using varying probe wavelengths and discovered that when we shifted to a larger wavelength, the bleaching signal transformed into an absorption signal, as illustrated in Fig. S25. This observation supports the deduction that small energy states were involved in the signal flip. Specifically, the bleaching signal originated from these small donor states, exhibiting similarities to the ground state bleaching signal. However, the larger wavelength was inadequate for exciting electrons within these states.

In light of these new experimental findings, our previous attribution of the increased bleaching signal to impact ionization now appears to be unreasonable. Nevertheless, the enduring bleaching signal continues to uphold our hypothesis that electron lifetime can be prolonged following modification with STO nanolayers.

Since this portion of the data does not provide additional insights beyond other characterization methods, such as TPV, we have relocated the MIR-TAS data to Fig. S25. Consequently, we have amended the description in the main text and the schematic related to "impact ionization" accordingly.

In lines 301, page 28, the original sentences of supplementary Information 'In Fig. S25a, both bare PTO and STO exhibited absorption signals in MIR-TAS results, while PTO-STO showed a long-lived bleaching signal. To investigate the formation of the

bleaching signal in the PTO-STO sample, the probe light wavelength was varied. As shown in Fig. S25b, the PTO-STO sample exhibited a bleaching signal at probe wavelengths of 3400 nm and 4000 nm. However, when the probe light wavelength was increased to longer than 6000 nm, the signals transitioned to absorption, resembling the shift from ground state bleaching (GSB) to normal absorption observed when the probe wavelength changes from shorter to longer. This indicated that interface states were formed during STO growth. Short-wavelength light could excite these interface states, resulting in a bleaching signal, whereas longer-wavelength light was insufficient to excite these states and thus showed absorption' were revised to description of the interface states after growth STO on PTO.

Fig. S25 a, MIR-TAS of PTO, STO, PTO-STO samples. **b**, MIR-TAS of PTO-STO sample with different probe light. **c**, Normalized MIR-TAS of PTO-STO sample with 3400 and 4000 nm probe light.

Reviewer #3 (Remarks to the Author):

This paper investigates the reason for low photocatalytic activity of ferroelectric nanoparticles, relating it to high levels of surface defects, showing that these can be reduced using a surface passivation layer of SrTiO₃. The central question of the paper why despite their benefits of enhanced charge separation via polarisation, do ferroelectrics still display very low photocatalytic activity – is a very pertinent one in the community. Answering this question will definitely be of benefit to many researchers working on ferroelectric photocatalysis and photocatalytic systems in general. While not giving evidence that this issue is a universal one affecting all photocatalysts, since only a single material is investigated, the evidence that passivation enhances photocatalytic activity in this case is compelling, and will likely prompt researchers to try similar approaches in different systems. The change in photocatalytic activity is backed up by detailed characterisation of the material's structural properties and charge-carrier dynamics, which help to support the conclusions that reduction in structural defects is the key driving factor in the enhanced activity. Certainly, a good correlation is demonstrated between STO coverage/morphology and performance, giving good evidence that it is the surface passivation, rather than STO contribution itself that is driving the enhancement. Finally the researchers report an impressive photocatalytic activity for a ferroelectric catalyst. Therefore, overall I believe the paper is worthy of publication, provided some questions are answered and revisions are addressed.

Q1. The most major factor that is not evidenced in the paper, is whether by adding the STO layer the eventual surface defects are almost completely eliminated, or whether they are just shifted to the STO surface, rather than PTO. The TEM work in the paper is excellent, and clearly shows a reduction in structural defects at the PTO/STO interface compared to the bare PTO surface. However, the authors have not included characterisation of the STO surface regions. This raises the question of whether the enhancement in activity is mainly because PTO is much less defected, therefore there is much less recombination in that layer, or also because the catalytic surface presented to the solution (STO in the composite case) is itself much less defected, leading to improved surface catalytic activity. To answer this question, the authors should include detailed TEM characterisation of the STO surface, as well as the PTO/STO interface and PTO surface. The TOF-SIMS in Figure 4d does look like there is some major change in composition around the surface, which suggests some reconstruction/defects. This again supports the need to characterise the STO surface in more detail.

Response:

Thanks for your valuable advice. We performed a TEM characterization of the STO surface, most of region of STO surface showed well structure, this less defective structure contributed to better activities. Additionally, there are only little region showed distortion, which showed the composition change in the surface.

STO growth on PTO still had some defects region, which could be seen in Fig. S14, and could also be verified by the EELS data in Fig. 4b. The reduced surface defects region greatly decreased surface recombination.

In lines 270, page 14, ‘Additionally, the decreased distortion regions in STO surface indicated less defects region in STO surface (Fig. S14)’ were revised to show the surface structure of STO.

Fig. S14 Surface structure of STO growth on PTO. **a,b**, HAADF-STEM and ABF-STEM data of STO surface.

Q2. Further, there are a number of minor points that should be addressed before publication:

1. Why are the particles monodomain? For 600 nm particles I would expect them to form multiple domains. The authors should explain/justify this more clearly in the manuscript.

Response:

The emergence of monodomain properties in PTO is theoretically plausible. Both theoretical predictions and experimental results support that PTO can exhibit monodomain characteristic.

Firstly, the monodomain properties of PTO could be justified by the comparison PFM data before and after annealed above the Curie temperature. The monodomain properties of PTO were experimentally confirmed through PFM results. The phase images shown in Fig. 1c and 1e indicate the absence of domain walls within the particles. Additionally, PFM measurements of PTO heated to 950°C (above the Curie temperature) revealed significant domain walls in Fig. S2.

Secondly, theoretically, the domain size can be described using the Landau-Lifshitz-Kittel (LLK) model, which considers factors such as the surface energy of boundary surfaces between domains, the electric field energy of the configuration, and the anisotropy energy of the polar orientation. (Reference: **Theory of the Structure of Ferromagnetic Domains in Films and Small Particles, Charles Kittel, *Phys. Rev.* 1946, 70, 965-971, Doi 10.1103/PhysRev.70.965**). When the energy is minimized, the relationship between domain width D and film thickness T is described by the following equation:

$$D \propto (\sigma_w T / E_p^2)^{\frac{1}{2}}$$

Where σ_w was the surface energy density and E_p was electric field energy in the polar domain. Typically, particles exhibit a larger domain size compared to films due to their higher surface energy. Additionally, polarization screening reduces the electric field energy, leading to an increase in domain size. In our case, the PTO particles, with their morphology and surface defects that screen the electric field, exhibit larger domain sizes than those reported for films. (**Fractal Dimension and Size Scaling of Domains in Thin Films of Multiferroic BiFeO₃, G. Catalan *et al.*, *Phys. Rev. Lett.*, 2008,**

Fig. 1 Characterization of monodomain PTO particles. **a**, SEM and HR-STEM of PTO particle. Figure 1, 2 and 3 are the corresponding HR-STEM of zones magnified as marked in the SEM image. **b,c**, Images of PTO with different polarization directions showing **(b)** amplitude and **(c)** phase of these two particles, respectively. Red line in **(b)** and blue line in **(c)** represent the values of the amplitude and phase of the two particles, respectively. **d**, PFM hysteresis loops of PTO particles. **e**, magnified phase image of a single domain PTO particle. (The scales of the bar: **(a)** 1nm, **(b)** and **(c)** 700 nm, **(e)** 240 nm, respectively).

Fig. S2 PFM data of PTO after annealed at 950 °C. **a**, PFM amplitude of PTO particle after annealed at 950 °C. **b**, PFM phase of PTO particle after annealed at 950 °C. **c**, amplitude (red line) and phase (blue line) of PTO particle align with the green arrow in **(a)** and **(b)**.

Q3. Introduction: “memorizer” should either be memristor, or memory.

revision:

We have revised the corresponding sentence in Line 62, Page 4: “One focuses on the polarization variation, such as ferroelectric **memorizer**,^{2,3} negative capacitance,^{4,5} and electrocaloric effect.^{6,7}”

Q4. “ferroelectrics are unique in that carriers mostly drift within the depolarization electric field”: in most cases depolarisation field is fully screened (otherwise it would depolarise!), so it is not the case that carriers mostly drift with the depolarisation field in ferroelectrics, as this has been screened. Unless the authors mean prior to screening. This point should be clarified/corrected.

Response:

Thanks for your comments. I agree your opinion that in mostly cases, a fully screened would be formed to make the polarization in ferroelectric stable. I want to

mean the charge could drift in the depolarization field before it be fully screened.

The revision in line 73, page 4 has been made: ferroelectrics are unique in that, prior to screening carriers mostly drift within the depolarization electric field.

Q5. In Figure 1d the phase difference between the plateau regions is only ~100 deg, not 180. This implies incorrect calibration and should be re-measured.

Response

Thanks for the correction. I have corrected the graph to make it clearer and more visualized. The phase in the graph was around -90° and 90° . The updated results can now be seen in Fig. 1c.

Fig. 1 Characterization of monodomain PTO particles. **d**, PFM hysteresis loops of PTO particles. **e**, magnified phase image of a single domain PTO particle.

Q6. P9 and Fig 2a,b: The description of what the red, blue and yellow dotted lines/regions denote is very hard to follow and understand. Particularly the phrase “are marked by blue and yellow dotted lines to denote bulk and surface regions, respectively” I cannot see how the blue and yellow dotted lines correspond to the bulk and surface regions, which are labelled on either side to the red dotted line. This should be made much clearer.

Response:

Thank you for your suggestions. Clearer results were showed in Fig. 2d-e.

I have made the data clearer by performing Fast Fourier Transform (FFT) for both bulk and surface regions. The difference in the (101) peak was evident in the FFT data. In the bulk region, the angle between the (101) peak and the X-axis was approximately 41.56° , while in the surface region, it shifted to 43.09° . This change indicated distortion in the surface region.

Fig. 2 Surface structure of PTO. d, STEM of PTO near positive polarization surface, the blue and yellow dot lines are the atom row of PTO in bulk and surface, respectively. The red dot line refers to the bulk and the surface bound. e,f, showed the FFT results of bulk and surface region, respectively. The circle represents the (101) direction of PTO, and the angle was the angle between the (101) point and the x-axis.

In line, 189, page 10, 'The angle of the (101) direction in the bulk region was

measured to be 41.56° relative to the x axis. However, at the surface, this angle shifted to 43.09° , suggesting the presence of distortion in the surface region. ” were revised to show the surface distortion in PTO.

Q7. In parts of the narrative the authors have not included enough information to fully understand what has been done or to explain their results. These should be explained more clearly, including:

a. P10: “This was achieved by selective growth of STO nanolayers on the PTO surface”

At least some detail should be included on how this selective growth was achieved, as this is very important to understand how the structural improvement of the PTO surface/interface may have been achieved. This is too important to just leave to an experimental section (which isn’t referred to anyway).

Response:

We added the descriptions and Fig. S8 in supporting material to describe the synthesis procedures:

In line 121, page 9 of Supplementary Information, we added: ‘The growth of STO on PTO was induced by the electrostatic adsorption of STO precursors onto the charged STO surface. As shown in Fig. S8, in the initial stage, negatively charged STO precursors were electrostatic adsorbed onto the positively polarized PTO facet, which contained positive bound charges. Following adsorption, the STO precursors grew into nanolayer films. This electrostatic adsorption mechanism was universal, as evidenced by the selective growth of BFO, CdS, TiO₂ and FeO_x as demonstrated in Fig. S8 b-e.’

Fig. S8 growth mechanism of STO on PTO. **a**, schematic of STO growing mechanism on PTO surface. **b**, BiFeO₃(BFO), **c**, CdS **d**, TiO₂, **e**, FeO_x selectively growth on PTO.

Q8. b. P11: “Consequently, the STO nanolayers selectively grown on the positive polarization facet led to negative polarization facet exposed” This is not easy to follow. The explanation should be made clearer.

Response:

We presented a detailed analysis to confirm the direction of polarization in PTO and to clarify the position of STO growth.

In line 220, page 11, ‘To identify which polar facet was exposed, a spherical aberration calibrated scanning transmission electron microscopy study was conducted. We selected the interface between red and blue rectangle regions in Fig. 3b, corresponding to the magnified region marked by red rectangle in Fig. S22. It is evident that the opposing side was not covered by STO’ were **revised** to explain the growth position of STO layers.

Fig. 3 Selective growth of STO on PTO and the charge separation image studies. **a,b**, SEM of PTO-STO structure. **c**, STEM of the interface between PTO and STO on the positive polarization facet. The picture at the top right is magnified at the red rectangle, where the blue, yellow, and red balls represent Pb, Ti, and Sr atoms, respectively.

Fig. S22 Surface structure of PTO-STO samples. **a**, the cross-section of PTO-STO. **b**, **c**, the region in (a) signed by red rectangle

Q9. c. P11: “Additionally, the results of photodeposition of Au and MnOx (Fig. S5) demonstrate consistent charge separation directions” The authors seem to be expecting us to read the SI to understand what they mean by this, but even the SI explanation is

not clear. This should be explained briefly in the main paper.

Response:

We have provided clearer images and revision of photodeposition in Fig. 2d-2f, illustrating the photodeposition reactions as follow:

In lin 241, page 12, ‘Ag⁺ obtained electrons and formed Ag, while Mn²⁺ obtained holes and formed higher valence state MnO_x. The deposition of Ag on the STO film (Fig. 3d) indicated that electrons separated to STO film, while deposition of MnO_x on uncovered PTO (Fig. 3e) showed holes transferred to exposed PTO surface. Furthermore, the co-deposition results indicated electrons and holes transferred to different direction (Fig. 3f)’ were **revised** to explain the photodeposition and charge separation.

Q10. d. P11: The authors briefly mention a Rh cocatalyst, but this is the first time it is mentioned and it is not discussed again until much later. So, they should explain the context of this more clearly. Also, again, they overly rely on the SI to explain what they are talking about, when the SI is not very clear either.

Response:

We have revised and added the following paragraph to the revised manuscript to confirm the position Rh where reduction reaction take place.

In line 363, page 18, ‘From the previous SEM images of the PTO-STO structure in Fig. 3b and Fig. S22, the STO on the positive polarization direction of PTO exhibited a

particle morphology, with gaps emerging between the STO particles. To determine whether the electrons separated to STO contributed to water splitting, the position of the hydrogen evolution cocatalyst, Rh, was investigated. Infrared (IR) spectroscopy results (Fig. S23) confirmed that Rh was located on the STO film rather than in the gaps between the STO particles and PTO.'

Q11. P12: "Since presence of the defective structure near the surface of PTO is confirmed, the nature of the defects should be further elucidated" The authors have just been talking about the PTO/STO system, where they say there are not any defects on the PTO surface. Here I assume they are talking about bare PTO (which is the case for Fig 4c), but this is not very clear.

Revision:

Yes, we want to talk about bare PTO, and we made a revision to make it clear:

'Since presence of the defective structure near the surface of bare PTO has been confirmed in Fig. 2, the nature of the defects should be further elucidated', in line 257, page 13.

Q12. Figure 6d: The text states that there is something showing the relaxation of hot carriers in the 'bottom left' of 6d. However, I can't see this. In general, 6d is very hard to follow/understand and should be improved.

Response:

Thanks for the advice, the similar questions have been raised by referee 1. We have made corrections to this part:

We attributed the bleaching signals to the impact of interfacial small energy states.

Upon re-conducting the experiment with different probe wavelengths, we observed that an increase in wavelength led to the transformation of the bleaching signal into an absorption signal, providing evidence for the existence of these small energy states. Additionally, the persistence of the long-lived bleaching signal further reinforced our hypothesis that electron lifetime can be extended through STO nanolayer modification.

We conducted a series of experiments using varying probe wavelengths and discovered that when we shifted to a larger wavelength, the bleaching signal transformed into an absorption signal, as illustrated in Fig. S25. This observation supports the deduction that small energy states were involved in the signal flip. Specifically, the bleaching signal originated from these small donor states, exhibiting similarities to the ground state bleaching signal. However, the larger wavelength was inadequate for exciting electrons within these states.

In light of these new experimental findings, our previous attribution of the increased bleaching signal to impact ionization now appears to be unreasonable. Nevertheless, the enduring bleaching signal continues to uphold our hypothesis that electron lifetime can be prolonged following modification with STO nanolayers.

Since this portion of the data does not provide additional insights beyond other characterization methods, such as TPV, we have relocated the MIR-TAS data to Fig. S25. Consequently, we have amended the description in the main text and the schematic related to "impact ionization" accordingly.

In lines 301, page 28, the original sentences of supplementary Information 'In Fig. S25a, both bare PTO and STO exhibited absorption signals in MIR-TAS results, while

PTO-STO showed a long-lived bleaching signal. To investigate the formation of the bleaching signal in the PTO-STO sample, the probe light wavelength was varied. As shown in Fig. S25b, the PTO-STO sample exhibited a bleaching signal at probe wavelengths of 3400 nm and 4000 nm. However, when the probe light wavelength was increased to longer than 6000 nm, the signals transitioned to absorption, resembling the shift from ground state bleaching (GSB) to normal absorption observed when the probe wavelength changes from shorter to longer. This indicated that interface states were formed during STO growth. Short-wavelength light could excite these interface states, resulting in a bleaching signal, whereas longer-wavelength light was insufficient to excite these states and thus showed absorption' were revised to description of the interface states after growth STO on PTO.

Fig. S25 a, MIR-TAS of PTO, STO, PTO-STO samples. **b**, MIR-TAS of PTO-STO sample with different probe light. **c**, Normalized MIR-TAS of PTO-STO sample with 3400 and 4000 nm probe light.

Q13. The decay curves in Fig S19 should be normalised so the kinetics can be more easily compared (probably show normalised plots as well as as-measured intensities).

Response:

We have added the normalized results and intensity of pump light (205 μJ) to the Fig.

S26.

Fig. S26 Slow TAS of different samples. **a**, the decay signal of different samples. Green dots on the line were when the $m\Delta OD$ decayed to half of the initial. **b**, normalized decay signal of different samples. **c**, Lifetime of different samples obtained from (a).

Q14. The SI figures should be re-ordered so they follow the order of the main text more logically.

Response:

Based on your and other reviewers' suggestions, we have re-ordered the SI figures to better align with the logical flow of the main text.

Reviewer #4 (Remarks to the Author):

Ferroelectric effects can potentially enhance solar energy conversion with photocatalysts. This manuscript evaluates this possibility for PbTiO₃ coated with SrTiO₃. The manuscript is interesting for the readership of Nature Comm. The quality of the Figures and of the text is very good. I recommend publication after a major revision:

Q1. The ferroelectric enhancement of overall water splitting is the main point of this work, but it is not directly demonstrated, only indirectly implied. The authors should include data to show that reversal of the polarization direction or heating above the Curie temperature will remove the ferroelectric enhancement of water splitting. See for example Assavachin, S. J.Am.Chem.Soc. 2023, 145(34), 18825-18833 and Morris, M.R., Advanced Materials, 2016, 28(33), 7123-7128.

Response:

We conducted the experiment as per your advice and found that the photocatalytic activity of PTO-STO significantly decreased after annealing it above the Curie temperature.

The result was added in Supporting Information part Fig. S21. And the following discussion was added also in lines 262, page 24, in supporting information: **The data on the photocatalytic activity of PTO-STO after heating above its Curie temperature is shown in Fig. S21. After heating PTO to 950 °C, the formation of multidomain**

structures was observed (as seen in Fig. S2), along with a reduction in the photocatalytic activity of PTO-STO to approximately half of its original level. This decrease in activity is attributed to the change in polarization direction, directly demonstrating the significant impact of ferroelectric properties on photocatalytic performance.

Fig. S21 Overall water splitting reactivity of different PTO-STO samples after annealed at 950 °C.

Q2. The suppression of defects by the SrTiO₃ layer is a central theme in this work, but the defects are not chemically identified in the text. Please include additional data on the nature of the defects and explain why they cause charge recombination.

Response:

Photoluminescence (PL) test was conducted the charge recombination through defects.

Photoluminescence (PL) data typically reflect the recombination between electrons

and holes. During process producing PL, electrons first capture by the defects, and then the captured electrons could recombine with trapped or free holes. The extra energy could be released at luminescence. After the growth of STO nanolayers, the PL intensity significantly decreased, indicating a reduction in recombination.

This **revision** was added to **Fig. S27** in page 30 of Supplementary Information with the description: **PL typically arises from the recombination of electrons and holes in semiconductors. The data showed that PTO-STO samples exhibited a spectrum similar to the background, whereas PTO samples displayed a distinct PL peak around**

Fig. S27 Photoluminescence (PL) spectrum of PTO before and after STO modification.

Q3. It is not clear if the charge carriers for water splitting are generated by excitation of the PbTiO_3 or by the SrTiO_3 . The latter is a well-known photocatalyst for overall water splitting. 100% quantum yield at 365 nm has been demonstrated. Takata, T., Nature, 2020, 581(7809), 411-414. Please include data to show PbTiO_3 is driving the

water splitting process, not SrTiO₃.

Response:

To exclude the contribution of STO in water splitting, a control experiment has been conducted that mixed PTO with STO where the amount of STO was the same as that growth in PTO. The activities of water splitting were significantly lower than PTO-STO we constructed. So, the water splitting was driven by PTO rather than STO.

Q4. At 360 nm illumination, why can charge carriers from SrTiO₃ be neglected?

Response:

To exclude any significant contribution of charge carriers from STO to water splitting, we conducted a controlled experiment, depicted in Fig. S21, as suggested by another reviewer. Our findings revealed that the photocatalytic activity notably diminished when PTO was transformed into a multidomain structure within the PTO-STO system. This outcome further substantiates that the ferroelectric properties of PTO are the primary drivers of the observed photocatalytic activity. Consequently, SrTiO₃'s role in charge generation is minimal; instead, it primarily serves to modify surface properties without substantially increasing carrier generation.

Q5. A band energy scheme with band edges should be shown to discuss the possibility of electron and hole transfer from PbTiO₃ to SrTiO₃.

Response:

The PTO-STO structure does not form a traditional heterojunction, STO mostly serve as modifier, so in our system, position of band edge was not provided.

The model can also be verified as evidenced by the comparison with the BTO-TiO₂ structure in the literature (N. V. Burbure, P. A. Salvador, G. S. Rohrer, *Chem. Mater.*, **2010**, *22*, 5823; A. Bhardwaj, N. V. Burbure, G. S. Rohrer et al., *Chem. Mater.*, **2010**, *22*, 3527.), as well as our own catalytic activity and SPV results. In our system, STO functions primarily as a modifier rather than a component forming a heterojunction. Therefore, charge transfer in this case does not follow the heterojunction model.

Q6. Fig 3f shows the SrTiO₃ only covering part of the PbTiO₃ crystal. Is this correct?

Please explain.

Response:

Yes, STO selectively grew on the PTO surface through electrostatic adsorption of negatively charged STO precursors onto the PTO facets, as verified by the SEM in Fig. 3a, Fig. S22, and EDX in Fig. S7. The growth mechanism is illustrated in Fig. S8. Based on these results, STO selectively grew on the positive polarization facet and the non-polar facet. A schematic cross-section of PTO-STO is provided in Fig. 3i (previously Fig. 3f) to clearly illustrate charge transfer within the PTO-STO particles.

Fig. 3 Selective growth of STO on PTO and the charge separation image studies. **a,b**, SEM of PTO-STO structure. **c**, STEM of the interface between PTO and STO on the positive polarization facet. The picture at the top right is magnified at the red rectangle, where the blue, yellow, and red balls represent Pb, Ti, and Sr atoms, respectively. (The scales of the bar: (a) (b) (d) and (e) 100 nm, (g) and (h) 800 nm, respectively).

Fig. 3a shows the structure of PTO-STO particles, where STO covers both the positive polarization facet and non-polar facet. This was determined through SEM and an analysis of the growth mechanism of STO on PTO. Briefly, STO nanolayers grew on PTO primarily due to the electrostatic adsorption of negatively charged STO precursors onto the PTO facets. The electrostatic repulsion between the negatively charged precursor and the negative polarization facet, which has a negative bound charge, hindered the adsorption and growth of STO on that facet. This growth mechanism is illustrated in Fig. S8, while the structure can be observed in the SEM

images in Fig. 3a and EDX data in Fig. S7.

Fig. S8 growth mechanism of STO on PTO. **a**, schematic of STO growing mechanism on PTO surface. **b**, BiFeO₃(BFO), **c**, CdS **d**, TiO₂, **e**, FeO_x selectively growth on PTO.

Fig. S7 EDS data of selective growth of STO on PTO. **a**, SEM image of PTO-STO. Images of the elements distribution are shown in (b) to (e): **b**, Ti, **c**, O, **d**, Sr, **e**, Pb, **f**, combination of Pb and Sr.

Q7. Masses and volumes for all reagents and solvents should be given to allow others to reproduce this work.

Response:

We made a revision that adds a detailed preparation process to the experiment section.

In line 533, page 32 the method of prepare PTO could be seen as follow: PbTiO_3 single domain crystals were prepared by hydrothermal method, following a previously established procedure. 0.2396 g Anatase TiO_2 (Alfa), 1.2421 g $\text{Pb}(\text{NO}_3)_2$ (Aladdin) and 3.36 g KOH (Aladdin) were added in 5 mL, 10 mL and 15 mL deionized water respectively. The resulting solutions were then transferred to a 50 mL Teflon-lined reactor, briefly stirred, and heated at 260 °C for 36 hours. After reaction, the yellow powders were washed 3 - 5 times with deionized water and dried at 80 °C.

In line 541, page 32 the method of prepare PTO-STO could be seen as follow: 2.4 g NaOH (Aladdin) was dissolved in 15 mL DI water, and then 3 mL 0.1 M tetrabutyltitanate(dissolved in 1 M HNO_3) was added in NaOH with stirring. After that, 12 mL DI containing 0.0635g $\text{Sr}(\text{NO}_3)_2$ was added to the mixture. Finally, 0.909 g prepared PbTiO_3 was added. After a 30 min stirring, the mixture was transferred to a 50 mL Teflon-lined reactor and heated at 200 °C for varying reaction times. After reaction, the resulting products were washed 3–5 times with a combination of ethanol and water and subsequently dried at 80 °C.

Q8. The term ‘highly efficient ‘should be replaced with quantitative information

Response:

We add the quantitative information in the description as follows:

In line 445, page 22, 'The defects-reduced structure resulted in the development of a highly efficient ferroelectric photocatalyst for overall water splitting producing H₂ 216.83 μmol/h and O₂ 116.9 μmol/h, achieving a remarkable apparent quantum yield of 4.08%' were **revised** to make it quantitative.

Q9. A reference needs to be added after '...established procedure.' On line 496

Response:

The revision has been made that added to this position.

Q10. The authors should say how H₂ and O₂ were measured. Was a closed loop reactor used? If not, how was air contamination accounted for?

Response:

The photocatalytic reactions were conducted using a closed-loop reactor system. Prior to testing, the reactor was evacuated using a vacuum pump to remove any residual air. During the reaction, the produced gases, H₂ and O₂, were analyzed using a gas chromatograph (GC). The GC chromatograms displayed peaks corresponding to H₂, O₂, and N₂. The amounts of H₂ and O₂ produced were quantified based on the areas of their respective peaks. Any minor air contamination was accounted for by the presence of N₂ peaks, which allowed for accurate deduction of the background contamination.

We made some revisions that added some details in the experiment section:

In line 560, page 33 the method of Photocatalysis process: The photocatalytic reaction was conducted by placing the high borosilicate glass reaction vessel, with a quartz cover and containing 100 mL of deionized water, in a water bath at 15 °C. The

reaction was irradiated with a 300 W Xe lamp. A total of 200 mg of the photocatalyst, with cocatalyst loaded *in situ* without further purification, was used. Sampling for product analysis via GC was performed at one-hour intervals. After each sampling, the reaction vessel was subjected to vacuum for 10–15 minutes. The entire reaction process was carried out using the Perfectlight Sci&Tech reactor. For quantifying H₂ and O₂ production, external standard quantification was employed. Different amounts of H₂ and O₂, in a 2:1 ratio, were successively injected into the reactor containing 100 mL of water. After allowing 30–60 minutes for equilibrium, the gas was injected into the GC with an Ar carrier. A series of data were acquired, and the plot of the injected amounts of H₂(O₂) versus the peak areas provided the reactor's calibration coefficient. During the test, the amounts of H₂ and O₂ production were calculated using this coefficient and the peak areas of H₂ and O₂.

Q11. How were catalysts purified after photodeposition of Rh, CrO_x, and CoOOH?

Response:

During the photocatalysis process, Rh, CrO_x, and CoOOH were deposited *in situ*, as described in the experiment section: they were sequentially deposited onto the photocatalyst without any additional purification steps. This is the standard step in typical photocatalysis measurement.

Unveiling Charge Utilization Mechanism in Ferroelectric for Water Splitting

We would like to express our sincere appreciation for the time and effort you have devoted to providing such valuable suggestions. We have thoroughly reviewed the comments from the reviewers and have made corresponding revisions in the manuscript. Below are our point-by-point responses to the comments, along with a detailed description of the changes implemented in the revised version.

Contents

Referee 1	Pages 3–5
Referee 2	Pages 6–6
Referee 3	Pages 6–8
Referee 4	Pages 8–13

REVIEWER COMMENTS

Reviewer #1 (Remarks to the Author):

The authors provide here a detailed rebuttal letter with additional new measurements. The revision changed the interpretation of several aspects (e.g., transient optical measurements) and to some extent clarified some aspects. Although the manuscript has been improved as a result of this revision (and I appreciate the hardwork of the authors), I still keep my original opinion of the work that the presented hypotheses are not yet backed up with solid evidence yet. The provided evidence for improved catalytic activity in PTO/STO nanoplates is subtle and not fully coherent in my view in the revised manuscript.

Response

We appreciate the reviewer's efforts to enhance the quality of our manuscript. To provide clearer clarification, we have revised the manuscript to delve into the potential role of a heterojunction mechanism within our system. By integrating these new insights with our existing data, we have strengthened our conclusions, rendering them more rigorous and scientifically sound.

Revision

In line 332, page 16: 'Although enhanced photocatalytic activity was observed following the growth of STO nanolayers on PTO, the precise mechanism underlying this improvement remained elusive. Previous findings had suggested that the deposition of STO nanolayers not only reduced surface defects in PTO but also promoted the formation of a transition interface between the two materials. Building on these observations, we propose two potential roles for the STO nanolayers in enhancing

photocatalytic activity: The STO nanolayers could form a heterojunction with PTO. This heterojunction might significantly contribute to the observed enhancement in photocatalytic activity by facilitating charge separation and transfer at the interface. In this scenario, charge generation would occur within the space-charge region (SCR) at the PTO/STO interface, where the depletion layer aids in separating electrons and holes under illumination. It is worth noting that, unlike conventional centrosymmetric semiconductors, the depolarization field in PTO can further assist in electron and hole separation; Alternatively, the STO nanolayers might primarily serve as a surface modification layer, altering the surface structure and properties of PTO. In this case, the charges involved in water splitting would be predominantly generated within the PTO particle itself. The STO layer would then enhance the surface charge dynamics within PTO, possibly by improving charge transport or reducing charge recombination. Charge separation in this scenario would primarily originate within PTO, driven by its ferroelectric depolarization field. To ascertain which of these mechanisms plays a dominant role in the enhanced photocatalytic activity observed, further investigation is required.'

The evidence to support our opinion about the surface modification instead of the heterojunction mechanism could be seen in following description in the manuscript:

1. In line 370, page18: Conventionally, an increased width of the SCR is expected to enhance charge separation and consequently reactivity. However, the reactivity exhibits a volcano-shaped pattern, suggesting a more complex relationship between SCR width and reactivity. It should be noted that the maximum thickness of STO nanolayers on PTO-STO is approximately 65 nm, which is significantly less than the typical width of the SCR in heterojunctions (typically 150-200 nm with slight doping).

2. In line 389, page 19: Additionally, after annealing above the Curie temperature, the activity was reduced by nearly half, which indicated the majority of carriers came from PTO rather than STO, and that charge separation was primarily driven by the ferroelectric depolarization field rather than the heterojunction structure.

In addition, we provided a more detailed description of the evolution of EELS peaks, offering a clearer explanation of the surface changes in PbTiO_3 after the growth of SrTiO_3 nanolayers.

Revision

In line 274, page 14. 'After the growth of STO nanolayers, the electronic structure in the bulk region of the sample remained similar to that of bare PTO, with distinct splitting between the t_{2g} and e_g peaks observable. In contrast, the splitting between the t_{2g} and e_g peaks was larger in bulk STO due to its cubic phase and higher symmetry. Interestingly, at the PTO surface where STO nanolayers were grown, the splitting between t_{2g} and e_g peaks resembled that observed in bulk STO, suggesting that the PTO surface region acquired some cubic-like characteristics. Moreover, the transition between t_{2g} and e_g peaks splitting from bulk to surface regions exhibited an opposite trend compared to bare PTO, as illustrated in Fig. S4 and previously discussed. This cubic-like structural feature was further confirmed by the repair of Ti defects and the substitution of Pb with Sr during STO growth, as seen in Fig. S16. For a clearer comparison of the structural changes at the PTO surface before and after the growth of STO nanolayer, we extracted and analyzed EELS data from the PTO surface before and after STO growth'.

Reviewer #2 (Remarks to the Author):

At this stage, the authors improved the analysis, which supports their claims.

I have no objection for the article publication at this step.

Reviewer #3 (Remarks to the Author):

I thank the authors very much for their response to my original comments. The majority of the comments have been fully addressed. In particular, the most important point (#1) has been addressed both in the response, and through the clarification in the text. There are just a couple of outstanding minor points that have not been fully addressed, or addressed incorrectly. Once these are corrected, I am happy for the paper to be accepted.

They are:

Q1. Q3 in rebuttal: this has not been correctly changed. The references refer to ferroelectric memories. The phase should be changed to ‘ferroelectric memories’ not ‘ferroelectric memorizers’.

Response

Thank you for the corrections. We have revised the corresponding sentence in line 62, page 4: “One focuses on the polarization variation, such as ferroelectric memories,^{2,3} negative capacitance,^{4,5} and electrocaloric effect.^{6,7”}

Q2. The authors have misunderstood my point, and appear to have manually shifted the y-axis of the phase signal. This should not be done, as a y-offset can be an important

feature in a PFM phase loop. However, this was in part because I made a mistake in my original comment, for which I apologise. I misread the axis, and the original phase difference was indeed close to 180 degrees. Therefore the authors should re-instate the original version of the figure.

Response

I have reverted the graph to its original version, and the updated results are now shown in Fig. 1d.

Fig. 1 Characterization of monodomain PTO particles. **d**, PFM hysteresis loops of PTO particles.

Q3. Q7 in rebuttal: where the authors have added the text “The growth of STO on PTO was induced by the electrostatic adsorption of STO precursors onto the charged STO surface.” Should this read “...onto the charged PTO surface”?

Response

We apologize for the imprecise description in the previous sentence. We

specifically refer to the bound charge at the surface when we mention 'charged surface.'
The inverted charged precursors were electrostatically adsorbed, creating an external screening effect. We have provided a more precise description of this process in the revision.

Revision

In line 122, page 9 of Supplementary Information, 'The growth of STO on PTO was induced by the electrostatic adsorption of STO precursors onto the PTO surface with positive bound charges' was **revised** to make a precise description.

Reviewer #4 (Remarks to the Author):

The authors have responded satisfactorily to most of my concerns. There are some residual concerns, that can be addressed in a minor revision.

1. OK

2. OK

3. Authors mention the results of control experiments with STO PTO mixtures, but the data is not shown.

Response:

The control experiment involving PTO and STO mixtures is presented in Figure 5a, and the relevant description can be found in line 324, page 16, as follows:

To exclude the specific contribution of STO nanolayers to photocatalytic water splitting reaction, a control experiment was conducted wherein PTO was mixed with STO in a 10:1 molar ratio (equivalent to the ratio of the raw materials during the growth

of STO nanolayers). The resulting H₂ production rate was approximately 20 μmol/h, and the AQY was 0.2%@365nm, which is significantly lower than that of PTO-STO samples. This discrepancy suggests that the boosted reactivity is due to the synergistic effect between PTO and STO.

Fig. 5 Overall water splitting activities with the modification of STO nanolayers. **a**, overall water splitting activities and AQY of PTO, PTO mixed with STO and PTO-STO samples.

4. We know from work by Rohrer and coworkers, that ferroelectric enhancements of SrTiO₃ can result from direct contact between a ferroelectric and SrTiO₃. See for example <http://dx.doi.org/10.1021/cm1018025> Mixing SrTiO₃ with PbTiO₃ particles does not correctly test the hypothesis that charge carriers can be generated by excitation of the SrTiO₃ coating.

Response:

Following your advice in the initial revision, we conducted an annealing experiment. After annealing, a multidomain structure formed in PbTiO_3 , and the water-splitting activity dropped to nearly half of its original level. This result directly demonstrates the critical role of ferroelectric properties in influencing photocatalytic performance.

Based on these results, we concluded that the photocatalytic performance was primarily driven by ferroelectric properties rather than the heterojunction effect. While we cannot entirely exclude the contribution of carriers generated in the heterojunction, we believed this contribution was minor compared to the influence of ferroelectric-driven charge separation.

5. No band energy scheme was provided. Adding such a scheme would greatly benefit the understanding of this work.

Response:

We have included a band energy scheme similar to that in the work by Rohrer and colleagues to illustrate how activity varies with different STO thicknesses and to further clarify our conclusions. Note that this diagram may not fully represent the interface region, as we observed a transitional PTO surface structure following STO modification. The revision was made in line 424, page 43 in Supplementary Information:

Revision

Fig. S34: Schematic of charge separation in various samples. **a**, Charge separation in bare PTO. **b**, Charge separation in PTO-STO samples with a thinner modification layer. **c**, Charge separation in PTO-STO samples with a thicker modification layer.

Fig. S34 illustrated charge separation across different samples. In bare PTO, as shown in Fig. S34a, electrons separated toward the positive polarization facet. However, when they reached the surface, significant surface recombination occurred, limiting the photocatalytic activity. After the growth of STO nanolayers, as shown in Fig. S34b, the elimination of surface defects allowed electrons to transfer more readily to the STO surface and participated in H_2 production. However, when the STO layer was much thicker, the extended travel distance for electrons led to increased charge recombination within the STO layer, decreasing the number of electrons available for water splitting.

6. OK

7. OK

8. OK

9. OK

10. Based on the revised text, it is still not clear if a closed-loop reactor was used or not.

For this type of experiment, a closed loop is needed to avoid false readings from air

contamination.

Response:

During the photocatalysis tests, we utilized a closed-loop reactor. We have made a revision in line 601, page 34, 'The reaction was in a closed-loop reactor and irradiated with a 300 W Xe lamp.,

11. OK

12. New query: Why is anodic photocorrosion (Pb^{2+} to Pb^{4+}) not a problem in this system?

Response:

We propose that the reacting holes primarily transfer to the oxidation cocatalyst and contribute to water oxidation. Water oxidation and photocorrosion are competing reactions; we tested a series of PTO-STO samples, as shown in Fig. S20. The data indicated that all samples remained stable during the reaction. Furthermore, the ratio of H_2 to O_2 was close to 2:1. If holes were participating in the photocorrosion reaction, we would expect to observe instability in the catalysts and a deviation from the 2:1 ratio between H_2 and O_2 . Based on these results, we conclude that anodic photocorrosion is not a significant issue in our system.

Fig. S20 Overall water splitting reactivity of different PTO-STO samples. **a**, PTO-STO-1, **b**, PTO-STO-4, **c**, PTO-STO-8, **d**, PTO-STO-12, **e**, PTO-STO-24, **f**, PTO-STO-48.

Unveiling Charge Utilization Mechanisms in Ferroelectric for Water Splitting

I am submitting the revised manuscript entitled “**Unveiling Charge Utilization Mechanisms in Ferroelectric for Water Splitting**” (No. NCOMMS-24-31509B) for your consideration of publication in Nature Communications. We are very pleased to hear that the reviewers felt that we have presented interesting and innovative findings. We want to extend our appreciation for taking the time and effort necessary to provide such guidance.

REVIEWER COMMENTS

Reviewer #1 (Remarks to the Author):

I don't have further comments.

Reviewer #4 (Remarks to the Author):

This looks good now. I recommend publication as is.